# Unpacking the multilingualism continuum: An investigation of language variety co-activation in simultaneous interpreters

**Laura Keller**[1]*, **Malte C. Viebahn**[2], **Alexis Hervais-Adelman**[3], **Kilian G. Seeber**[1]

**1** Interpreting Department, Faculty of Translation and Interpreting, University of Geneva, Geneva, Switzerland, **2** Department of Psychology, University of Leipzig, Leipzig, Germany, **3** Institute of Psychology, University of Zurich, Zurich, Switzerland

* Laura.Keller@unige.ch, laura_keller@gmx.ch

## Abstract

This study examines the phonological co-activation of a task-irrelevant language variety in mono- and bivarietal speakers of German with and without simultaneous interpreting (SI) experience during German comprehension and production. Assuming that language varieties in bivarietal speakers are co-activated analogously to the co-activation observed in bilinguals, the hypothesis was tested in the Visual World paradigm. Bivarietalism and SI experience were expected to affect co-activation, as bivarietalism requires communication-context based language-variety selection, while SI hinges on concurrent comprehension and production in two languages; task type was not expected to affect co-activation as previous evidence suggests the phenomenon occurs during comprehension and production. Sixty-four native speakers of German participated in an eye-tracking study and completed a comprehension and a production task. Half of the participants were trained interpreters and half of each sub-group were also speakers of Swiss German (i.e., bivarietal speakers). For comprehension, a growth-curve analysis of fixation proportions on phonological competitors revealed cross-variety co-activation, corroborating the hypothesis that co-activation in bivarietals' minds bears similar traits to language co-activation in multilingual minds. Conversely, co-activation differences were not attributable to SI experience, but rather to differences in language-variety use. Contrary to expectations, no evidence for phonological competition was found for either same- nor cross-variety competitors in either production task (interpreting- and word-naming variety). While phonological co-activation during production cannot be excluded based on our data, exploring the effects of additional demands involved in a production task hinging on a language-transfer component (oral translation from English to Standard German) merit further exploration in the light of a more nuanced understanding of the complexity of the SI task.

## Introduction

Multilingual speakers' languages have been found to be simultaneously activated to varying degrees at all times [1–6], paving the way for investigations into multivarietal language

**Data Availability Statement:** Data cannot be shared publicly via a depository because of Faculty rules as applicable at the time of Ethics Committee approval (2016). A minimal data set (see

supporting information S2_file.xlsx) has been submitted with the revised manuscript to conform with journal regulations.

**Funding:** AHA is supported by the Swiss National Science Foundation, https://www.snf.ch/en, grant number PP00P1_163726. The funders had no role in study design, data collection and analysis, decision to publish, or preparation of the manuscript.

**Competing interests:** The authors have declared that no competing interests exist.

processing. Interestingly, empirical evidence suggests that language-variety processing follows multilingual language processing and that political or social considerations of what constitutes a language as opposed to a variety within a language appear to have little bearing on processing [7, 8]. Constant co-activation also does not seem to be at odds with precise and deliberate output-language selection, which conditions smooth oral communication.

This applies in particular to the context of simultaneous interpreting (SI). Simultaneous interpreters are required to comprehend a message in one language–the original–and render it in a different language in real time for the benefit of the listeners who do not understand the original. To complete this task successfully, both languages required to convey the information must be co-activated in the interpreters' minds for the message to be transferred from the source to the target language. All the while, interpreters must avoid production interference and assure that both content and linguistic form of the output meet the expectations in terms of accuracy and idiomatic expression. Interpreters' output-language selection is, therefore, not only conditioned by the task of transferring a message across languages; the context may require the use of a specific language variety or register to render the message faithfully. The time lag between input perception and output production must be small–a few seconds at most [9]–for speakers of different languages to effectively communicate across language barriers in real time. SI thus requires continuous comprehension and production in two languages [10]. As comprehension and production are usually part of discrete processing stages, SI constitutes a form of extreme language processing [11–20], and some of the sub-processes may increase in automation with greater expertise.

Activating and employing the appropriate sets of vocabulary, grammar and phonological rules at the right time out of many available options may increase the demand for competition resolution and affect the availability of resources [11, 21–24]. And not only for users of distinct languages: Evidence suggests that actively using language sub-varieties such as dialects may affect language processing similarly to using multiple languages [7, 8, 25–29]. From a processing perspective, there may therefore be little difference between bivarietal speakers and bilinguals.

## Implications of bivarietalism

Bivarietalism is a widespread experience, and if bivarietal and bilingual processing bear important similarities, a more nuanced understanding of its implications could help expand our understanding of bilingual language processing.

The focus of this study lies on bivarietal speakers of German in Switzerland who present an interesting test case. In German-speaking Switzerland, two varieties of German are habitually used: Standard German primarily for written communication and in very formal contexts and the dialectal variety of Swiss German for oral communication in all other contexts [30–33]. Swiss German is distinct from Standard German primarily on a phonological level [34] and is used independently of speakers' socio-economic status or level of education [35]. Because of the functional separation in variety use, German-speaking Switzerland was defined as one of the archetypical examples for diglossia [36]. However, as the focus of this study lies on the processing of language varieties in individuals rather than the socio-cultural realities of language [37], we refer to speakers of both varieties as bivarietal speakers, suggesting they present an extension to the concept of bilinguals.

As for bilingual language processing, bivarietal language processing is not straightforward. Bivarietal speakers (here speakers of Dundonian/Standard English and Öcher Platt/Standard German) show patterns comparable to those observed in bilinguals switching between languages when asked to complete a switching task involving a dialectal variety and a Standard,

i.e. the naming latencies were higher on switch than non-switch trials, and the switch costs in balanced bivarietals remained the same independently of the switching direction, while unbalanced bivarietals showed asymmetrical naming latencies, taking more time to switch from their non-dominant into their dominant variety than vice-versa [7]. In contrast, a picture-word interference task involving the same language-variety pair revealed no evidence for variety separation in the mental lexicon [38].

The patterns emerging from empirical evidence on bivarietal and bilingual processing therefore appear to be similar, with some of the data pointing to a separation in the lexicon [39, 40], supporting a hierarchical view [41, 42], and some of it favouring an integrated connectionist view [43–48]. We therefore propose to investigate bivarietalism as a form of bilingualism.

## SI experience and bivarietal co-activation

In SI, comprehension and production in two different languages overlap, as interpreters continue to comprehend the incoming speech stream while formulating the previously comprehended parts of the ongoing discourse [14, 20, 49, 50]. While simultaneous interpreters verbalise an external train of thought presented at a speed they do not control, like any other speaker they attend to their own speech stream for the purpose of output monitoring [51]. Given the complexity of the SI task, the many concurrent sub-processes it requires, and the cognitive demands this involves [18, 52, 53], it is conceivable that fewer cognitive resources are available for other processes such as prediction, sentence planning or production [10].

SI requires co-activation of the input and output language as both languages are relevant to the task. However, evidence from outside the SI context suggests that languages that are not task relevant are also co-activated [1–4, 54, cf. 55]. For a better understanding of the processes involved in the SI task, it is therefore of interest to investigate whether the activation of a task-irrelevant language variety can be measured and whether task type and habitual language use (here SI expertise) impact co-activation.

Bivarietal conference interpreters are a particularly pertinent sub-population to study as they speak a formal variety along with a dialectal variety they rarely use for output production in an SI situation–the contexts in which SI is provided generally requires the use of Standard German.

In summary, two premises frame this study: First, language co-activation as observed in multilinguals indicates that a language that is not used in a specific communication context is still being processed and is therefore at least partially active. Second, both interpreters and untrained multilinguals and bivarietal speakers make precise selections regarding language varieties, including register (e.g., lexical and syntactical choices). Additionally, as during comprehension, the selection process during speech planning may also be subject to cross-linguistic phonological competition [e.g., 56]. This leads us to the following questions: Does co-activation extend to what we consider to be language subsystems or language varieties? If so, does it vary depending on the nature of the task and on how speakers habitually use their languages and varieties, here specifically on whether or not they have expertise in SI?

## The present study

In a monolingual context, co-activation reflects the organisation of the multilingual lexicon, but does not otherwise serve an immediate obvious purpose. In an interpreting context, however, co-activation is a necessity, as the interpreters' comprehension system is engaged in input-language comprehension, while the production system is occupied with output-language production [10]. Against this backdrop, we designed a study to investigate whether

evidence for the co-activation of a task-irrelevant variety can be found and, in a second step, whether task type and SI skills have an incidence on language co-activation including language-variety co-activation. In the present study involving a language variety without a standardised written form (Swiss German), we investigate variety co-activation in the Visual World Paradigm (VWP), as it is particularly suited to covertly investigate word-form activation using object images only and entirely circumvents the explicit use of the task-irrelevant language variety. The VWP [for comprehensive reviews see 57, 58] has proven useful to examine language co-activation in multilinguals during comprehension [1–4] as well as during production [57, 59]. The effects of bivarietalism on cognition have previously been investigated [8, 25, 26, 28, 29], and the VWP has also been used to study predictive processing based on morphological structures in dialect users: Stable dialect speakers of the grammatically more specific variety of Norwegian make accurate predictions based on gender markings that differ between the varieties, while unstable dialect speakers and speakers of the grammatically less specific variety do not [60]. Stable dialect speakers therefore exhibit the same performance patterns observed in proficient bilinguals. Furthermore, owing to its minimal invasiveness, the VWP allows to simultaneously accommodate the specificities of the SI task (e.g., parallel processing of audio and visual input, the possibility of comprehending an utterance in one language and articulating its semantic equivalent in another) and those of language-variety processing.

The VWP was used to test whether empirical evidence for language variety co-activation can be found, hypothesising that if language varieties are processed like different languages, bivarietal speakers would show patterns comparable to bilinguals in the studies we based ours on, namely a significantly higher fixation proportion on the cross-variety competitor than on the distractors [1–6]. Furthermore, we set out to test whether task (comprehension vs. production) and SI related profile differences (interpreters vs. non-interpreters) affect co-activation levels, expecting lower levels of co-activation in production than comprehension, based on the assumption that the comprehension task is cognitively less challenging than the task combining a comprehension component in English with a production component of the translation equivalent in German, as well as lower levels of co-activation in interpreters than in non-interpreters, based on the hypothesis that experience with SI may lower the degree of co-activation of task-irrelevant varieties due to the high cognitive demands of the primary task of SI [11–20, 52, 53].

## Materials and methods

### Participants

64 native German speakers highly proficient in English and French participated in the study. 32 were conference interpreters trained in SI (age $M = 44.3$, $SD = 12.6$; 24 women), 32 had no SI training (age $M = 38.0$, $SD = 11.6$; 24 women). In addition, 16 members of each group were also speakers of Swiss German (i.e., bivarietal). All participants gave written informed consent (see S5 Appendix for the informed consent form used). The study received approval by the institution's ethics committee and was conducted in observation of the Declaration of Helsinki.

Requirements for recruitment were a native level of German and high proficiency in English and French for all participants; the bivarietal speakers had to have a Swiss German family background or have spent time in a Swiss-German speaking environment while in school (6+ years in a Swiss German speaking environment prior to turning 18). Overall, participants reported knowing between three and nine languages ($M = 4.5$, $SD = 1.2$, $Mdn = 4$), including Arabic (1), Catalan (1), Czech (1), Dutch (2), Farsi (1), Finnish (1), Hungarian (1), Italian (24), Japanese (1), Mandarin Chinese (1), Portuguese (7), Slovak (1), Spanish (30), Swedish (2), Romansh (1), Russian (5). The interpreters averaged 16.1 years of professional

**Table 1. Participants' biographical data and language background information.** Overview of the LEAP-Q analysis data (proficiency score 0 = none, 5 = 100%).

| | Bivarietal interpreters | Monovarietal Interpreters | Bivarietal non-interpreters | Monovarietal non-interpreters | *sign. differences* |
|---|---|---|---|---|---|
| *General* | | | | | |
| N | 16 | 16 | 16 | 16 | - |
| Age—mean (*SD*) | 44.8 (*13.6*) | 43.8 (*12.0*) | 33 (*9.8*) | 43.1 (*11.2*) | *G3 ≠ G1, G2, G4* |
| Age—range | 28–66 | 24–64 | 23–56 | 29–66 | |
| Gender—F:M | 13:3 | 11:5 | 13:3 | 13:3 | - |
| Handedness—RH:LH:AM | 14:1:1 | 16:0:0 | 15:1:0 | 15:1:0 | - |
| Years of education—mean (*SD*) | 16.6 (*3.6*) | 18.6 (*2.3*) | 16.8 (*2.7*) | 16.5 (*5.0*) | - |
| Languages spoken—mean (*SD*) | 4.7 (*1.0*) | 5.1 (*0.8*) | 4.6 (*1.4*) | 3.7 (*1.0*) | *G2 ≠ G4* |
| *Languages* | | | | | |
| **Standard German*** | | | | | |
| proficiency (speaking & listening) | 4.9 (*0.2*) | 5 (*0*) | 5 (*0*) | 5 (*0*) | - |
| acquisition onset (age, yrs) | 1.2 (*1.3*) | 0.6 (*0.8*) | 1.1 (*1.0*) | 0.8 (*0.9*) | - |
| fluency onset (age, yrs) | 3.9 (*2.7*) | 2.9 (*1.5*) | 4 (*1.3*) | 3.4 (*2.3*) | *G2 ≠ G3* |
| current exposure (%) | 39.9 (*22.1*) | 38.8 (*16.4*) | 36.6 (*14.8*) | 38.7 (*17.9*) | - |
| **English** | | | | | |
| proficiency (speaking & listening) | 4.5 (*0.7*) | 4.2 (*0.6*) | 4.0 (*0.6*) | 4.6 (*0.5*) | - |
| acquisition onset (age, yrs) | 12.0 (*3.5*) | 9.25 (*2.4*) | 12.1 (*2.1*) | 10.9 (*1.9*) | *G2 ≠ G1, G3, G4* |
| fluency onset (age, yrs) | 18.0 (*6.5*) | 15.2 (*2.8*) | 15.2 (*2.3*) | 18.7 (*6.1*) | - |
| current exposure (%) | 19.1 (*14.6*) | 17.7 (*9.3*) | 17.2 (*9.2*) | 20.5 (*15.9*) | - |
| **French** | | | | | |
| proficiency (speaking & listening) | 4.5 (*0.7*) | 4.4 (*0.6*) | 4.5 (*0.6*) | 4.4 (*0.7*) | - |
| acquisition onset (age, yrs) | 9.0 (*4.8*) | 12.4 (*5.1*) | 10.6 (*2.5*) | 11.4 (*5.7*) | - |
| fluency onset (age, yrs) | 16.1 (*6.1*) | 20.5 (*9.2*) | 17.6 (*4.3*) | 20.9 (*11.4*) | - |
| current exposure (%) | 33.8 (*24.9*) | 24.1 (*16.2*) | 35.8 (*16.3*) | 35.8 (*20.8*) | *G2 ≠ G1, G3, G4* |

experience (*SD* = 12.3) and all participants reported regularly using English and French at work. Based on their language profile and professional background, participants were assigned to one of four groups: Bivarietal interpreters, Monovarietal interpreters, Bivarietal non-interpreters or Monovarietal non-interpreters. They reported normal or corrected-to-normal vision and no history of speech, hearing or learning disorders. Table 1 provides an overview of the participant groups' language biographical profiles.

Data on Swiss German is not included in Table 1 as the participants filled in the language-background questionnaire before coming into the lab for testing and the implication of Swiss German was only disclosed to them once the experiment completed. However, all bivarietal participants completed a post-hoc naming task, to make sure that the cross-variety competitors were recognised as such. Due to recruitment restrictions, group 3 diverged from the other groups in age and self-rated English competence. The differences in age, German fluency onset and English acquisition onset were deemed potentially relevant to the analysis of the data. These factors were therefore included as fixed effects in the respective analysis models (see S3 Appendix). The corresponding analyses revealed no significant effect of the between-group differences on the results (see Results).

## Apparatus

The experimental tasks were completed in an ISO-compliant mobile interpreting booth (ISO 4043–2016) equipped with a Bosch interpreting console (DCN-IDESK-D). Eye-movement

data were acquired with an SR Research EyeLink® 1000 desktop-mounted remote eye-tracking system with a monocular sampling rate of 500 Hz from participants' dominant eye. Calibration was conducted using a nine-point grid. Visual stimuli were presented on a 19-inch ViewSonic CRT monitor with a refresh rate of 115Hz, located approximately 75 cm from the participants. The eye-tracker camera was positioned in front of the monitor, leaving approximately 60 cm between participants' eyes and the eye-tracker lens.

## Language background questionnaire

An adapted electronic version of the Leap-Q [61] was used to gather participant-profile data. The Leap-Q was adapted in that the Likert-like scales employed were limited to 5 points (vs. 10 in the original questionnaire) to allow the highly multilingual participants to limit the time spent on filling it in for a multitude of languages.

## Experimental design

The experiment comprised two tasks, one intended to study language variety co-activation during monolingual comprehension (the comprehension task) and one to investigate language variety co-activation during production in a multilingual setting (the production task). Task order was counterbalanced between the participants of each group to prevent effects of task order and visual priming.

**Task 1: The comprehension task.** The monolingual comprehension task aimed at investigating language-variety co-activation patterns during comprehension in trained interpreters and untrained multilinguals by measuring co-activation of the task-irrelevant language sub-system [analogously to 1–4, 6]. Participants were asked to click on a visual target displayed on a screen along with a phonological competitor and two distractor images (three distractors and no competitor in the baseline condition). Only Standard German was overtly used for this task. In the same-variety competitor condition, the target object name shared its phonological onset with the competitor name in Standard German, e.g., *Bus* (bus)–*Buch* (book). In the cross-variety competitor condition, the target and competitor did not share their phonological onset in Standard German, but the Swiss German name of the competitor was a cohort competitor to the target, e.g., *Haar* (hair)–*Haag* (fence; *Zaun* in Standard German; see Fig 1). The baseline condition contained no phonological competitors to the target in either of the varieties of German investigated. A randomized 3x2x2 mixed factorial design was used, with a within-subjects repeated measure (*competitor type*, 3 levels: Swiss German competitor, Standard German competitor, no competitor) and two between-subject factors (*interpreter status*, 2 levels: interpreters, non-interpreters and *variety status*, 2 levels: bivarietal, monovarietal speakers).

The 75 stimulus sets (25 per condition) used in the 75 critical trials participants completed contained the recurrent Standard German instructions (*Bitte klicken Sie auf*–Please click on), the spoken target and four black-and-white line object drawings. Every set was made up of a Standard German target, a phonological competitor (cross-variety or same-variety) and two unrelated distractors, or three unrelated distractors for the baseline. The placement of target and competitor was randomised and counterbalanced. Only the target names were used for audio input, neither the object names of the competitors nor of the distractors were heard during the experiment in either variety.

The target-competitor pairs were selected exclusively based on their shared phonological onset. Every stimulus set was compiled to avoid semantic relations and, to the extent possible, visual similarities among all four objects. To avoid unwanted phonological competition, the English and French equivalents of the distractor names were checked for word-onset

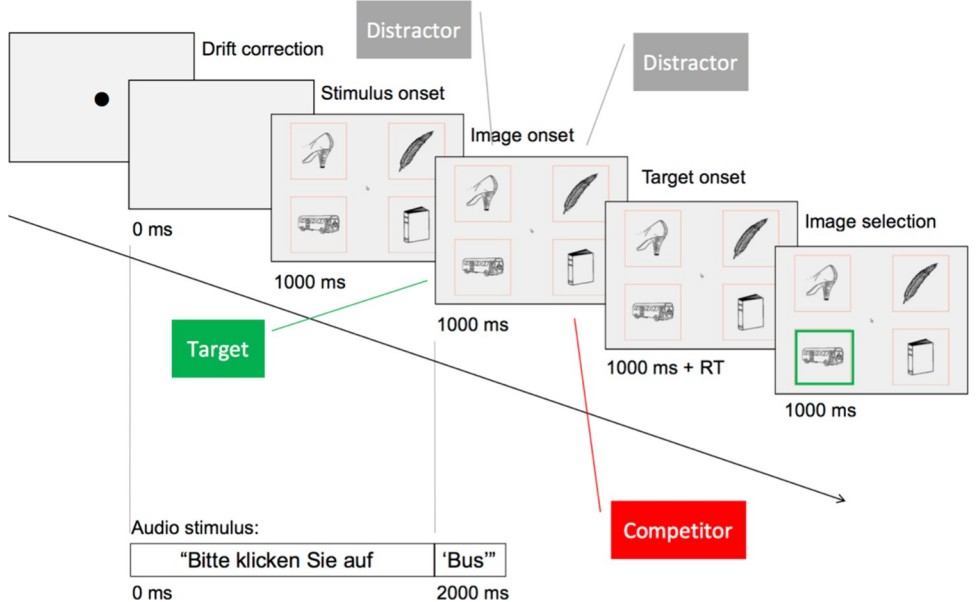

**Fig 1. Comprehension-task same-variety competitor trial with timeline.**

phonological overlap with the other objects in German, English and French. Furthermore, for the Swiss German competitors, only widely used terms without a strong regional affiliation, verified with the *Schweizerisches Idiotikon* dictionary of Swiss German, were used. For the Standard German competitors, the target slot was assigned to the object name with more synonyms, that is to the object presenting multiple rather than just one plausible option, reserving the single-option spot for the phonological competitor (see S1 Appendix for the full stimulus set).

The black-and-white line drawings for the visual stimulus component were taken from standardised databases [62–64] and completed with additional drawings created using the Laminar Pro Image Editor iPhone app (2016) where stimuli were not otherwise available. All images were manually edited in Adobe Photoshop CS6 to create contiguous lines and adjust for size (168x168 pixels, which corresponded to the outline of the interest area defined for each image, the parts of the screen that were not covered by the four interest areas containing the object images were defined as 'other' and excluded from the analysis) and file format (.png, transparent background). The images were positioned on the screen so that the inner corner of each area of interest was 2.85 degrees of visual angle away from the screen's midpoint.

The mean phonological overlap between same-variety and cross-variety target-competitor pairs for each set was 2.6 phonemes (*SD* = 0.67). Independent t-tests showed no significant difference across conditions (Swiss German competitor condition: *M* = 2.72 (*SD* = 0.79); Standard German competitor condition: *M* = 2.48 (*SD* = 0.51), *p* > .05, Cohen's D = 0.36; Baseline condition: no phonological overlap).

Word frequency was controlled [65] and frequency calculations were based on the University of Leipzig German language corpora collection [66]. As the competitors used in the cross-variety condition are part of a non-standard variety mainly used for oral communication without a written code, systematic frequency indications for Swiss German terms are unavailable. The measures used to calculate frequency for that condition were those for the Standard German terms referring to the same concept. While they only allow for an approximate measure of the frequency of the Swiss German equivalents of the Standard German words, this approach allows us to exclude that the objects used as Swiss German competitors appeal to the

monovarietal participants for reasons of higher frequency in Standard German. Target and competitor frequency were comparable within condition (condition 1: target $M$ = 13.48 ($SD$ = 2.58) vs. competitor $M$ = 14.32 ($SD$ = 2.39), $p$ > .05, Cohen's D = 0.34; condition 2: target $M$ = 12.44 ($SD$ = 2.27) vs. competitor $M$ = 12.32 ($SD$ = 1.99), $p$ > .05, Cohen's D = 0.06); relative target frequency was comparable across all three conditions (condition 1: $M$ = 13.48 ($SD$ = 2.58); condition 2: $M$ = 12.44 ($SD$ = 2.27); condition 3: $M$ = 12.56 ($SD$ = 2.60), p > .05, and Cohen's D = 0.42, 0.35 and 0.05, respectively for between-condition t-test comparisons).

Participants' dominant eye was determined with a sighting test. Table height and camera angle were adjusted to each participant, and a reference point sticker was placed centrally on their forehead, above the nasion and slightly above eye-brow level. Target-word onset systematically followed 2000 ms after onset of the instructions and 1000 ms after image onset. At the launch of each trial, participants were presented with a blank screen while hearing the first part of the instructions ('Bitte klicken Sie auf. . .'–'Please click on. . .'). 1000 ms before target-word onset, the trial set images appeared in the quadrants and the mouse cursor at the centre of the screen. Participants were instructed to click on the identified target object as quickly and accurately as possible. The instructions contained no article to avoid predictive processing based on grammatical gender marking in German.

The images remained visible until participants selected one of them, triggering visual feedback (a green frame around the selected image for correct, a red frame for incorrect), and disappeared again after 3000 ms. 1000 ms after image selection a blank screen with a new drift-correction point appeared, signalling the start of a new trial. To proceed, participants validated the drift correction by pressing the space bar. After completing a practice-trial session, participants launched the critical trials by pressing the space bar. Trials were displayed in a semi-randomized order, with no more than two trials per condition in sequence.

**Task 2: The production task.**   For deliberate multilingual communication and particularly to transpose a message from input (source) to output (target) language as in SI, production is a crucial process component. Evidence has been found for language co-activation also during production [57, 59, 67–69]. To investigate whether language variety co-activation could be observed in a translation situation, that is, when transferring a message from an input to an output language of which two varieties are available, the stimuli for the comprehension task were re-used. The production-task design differed from the comprehension task in that participants heard the English translation of the target embedded in an English sentence (the participants' L2) and were asked to perform a language-transfer task that included naming the target in Standard German, instead of clicking on the target image. Interpreters specifically were asked to simultaneously interpret the short English sentences into Standard German and to start production as quickly as they could to minimize the lag between in- and output, thus ensuring the partial overlap of comprehension and production typical for SI [9]. To avoid presenting non-interpreters with a disproportionately difficult task–simultaneously interpreting a non-sensical sentence was deemed very challenging for participants without training–and as the critical component of the task was accessing the mental lexicon and articulating the target word, they received the same input as the interpreters but were asked to only translate the sentence-final target word into Standard German. Participants articulated their verbal response according to the instructions their group received, and their responses were recorded via the microphone on the interpreter console installed in the test booth. Due to the task difference, the results for the two tasks were analysed separately.

The visual stimuli for both tasks were identical. The audio input for the production task consisted of English sentences with the target word in sentence-final (object) position, recorded by a female native speaker of Canadian English. In both tasks, onset of the target word in the audio stimulus came 2000 ms after stimulus onset and 1000 ms after image onset.

The structure of the English sentences was the same throughout: "The [subject] [transitive verb in past tense] the [target]", e.g., The scientist mentioned the traffic light (see S2 Appendix for the complete sentence lists).

Every agent (subject) and verb were used in one sentence per condition; the subject-verb combination differed across conditions, e.g., the subject 'scientist' was used in combination with the verbs 'to mention', 'to look at' and 'to move towards' over the course of the task. To encourage phonological co-activation, the sentences were crafted to avoid context-based top-down prediction and semantic co-activation. To further minimise sentence-context effects, a second sentence list was created by moving the targets allotted to the agent-verb combinations down one slot. Sentence-list attribution was counterbalanced between participants.

Varying the agent-verb combination aimed at increasing the ecological validity of the audio input. Interpreting an identical sentence onset over the whole experiment would likely have led to a slump in sustained attention.

To evaluate how much sense the sentences made, the lists were rated by 12 anonymous English native speakers (6 each; ratings from 1 = "makes no sense" to 6 = "makes perfect sense"). Sentence List 1 received a mean 'sensibility' score of 4.2 ($SD$ = 1.0), sentence List 2 of 3.6 ($SD$ = 1.4). Inter-rater reliability for both sentence lists was slight; Fleiss' κ for list 1 = 0.07, Fleiss' κ for list 2 = 0.13.

The production-task procedure and the visual materials used were identical in the comprehension and production tasks, whereby the objects were placed in different quadrants for each exposure. The procedure for the production task differed from the comprehension-task procedure in terms of audio input (target named in English and embedded in an English sentence) and task instructions (interpreters: 'simultaneously interpret the whole sentence' for; non-interpreters: 'translate the sentence-final word into Standard German). Fig 2 illustrates a production L1-competitor trial.

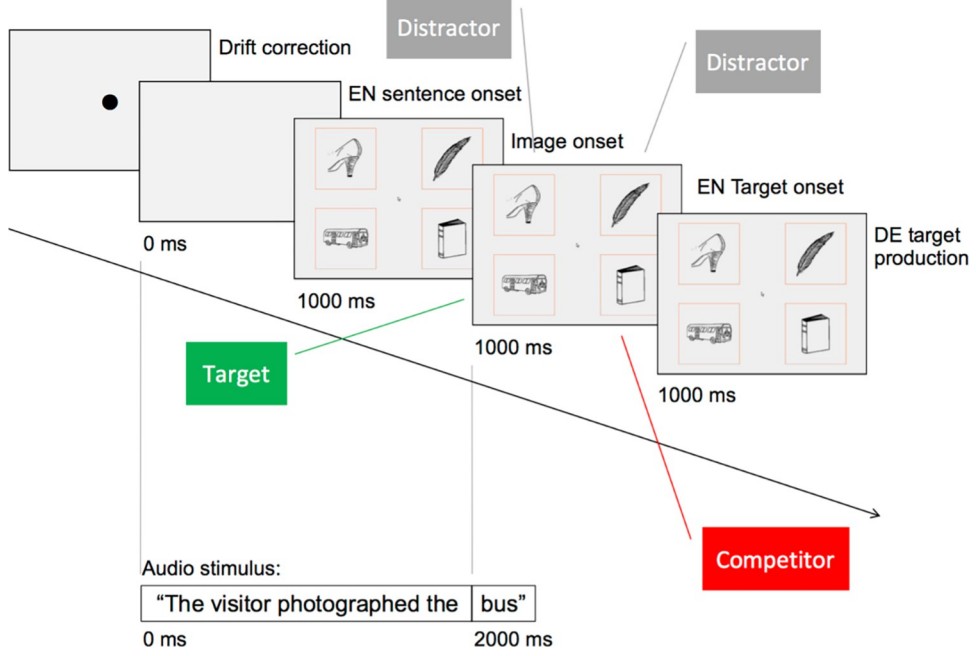

**Fig 2. A production-task same-variety competitor trial with timeline.**

## Data analysis

Response accuracy, reaction time (RT) and fixation proportion were analysed separately as dependent variables. Inferential analyses were carried out in R [70] using the *lme4* package [71]: linear mixed-effects models (LMM) were used for RT data, generalised linear mixed models (GLMM) for accuracy data, and growth curve analyses (GCA) for fixation proportion data. Inaccurate trials were excluded for mean fixation proportion analyses.

Data trimming was performed prior to RT and accuracy analyses. Following a conservative approach to outlier elimination, responses above and below 3 SDs from the mean were considered outliers and excluded.

To test the relationship between log-transformed RTs of accurate trials and the fixed effects, RT data were analysed using an LMM with the same fixed-effects structure as the GLMM above. Subjects and items were entered as random effects with by-subject and by-item random slopes for the effect of group membership, and by-subject and by-item random intercepts. As for the GCAs, for LMMs estimates ($\beta$), standard errors (SE), and t-statistics are reported, and the significance of effects was determined by assessing whether the associated t-statistics had absolute values of $\geq 2$.

To test the relationship between accuracy and the fixed effects *interpreter status* (2 levels: interpreters; non-interpreters), *language-variety status* (2 levels: bivarietals; monovarietals) and *condition* (competitor type, 3 levels: Swiss German competitor; Standard German competitor; no competitor) GLMM-accuracy analyses were conducted using the *glmer* function [72, 73]. Accuracy was entered as a binary variable. The binomial link family was set to logit. To test whether SI experience and bivarietalism interacted with each other, an interaction term was set between *interpreter status* and *variety status*. Subjects and items were entered as random effects with by-subject and by-item random intercepts (the maximal random structure supported by the data). For GLMMs, estimates ($\beta$), standard errors (SE), as well as Z-statistics and p-values are reported. P-values $\leq .05$ were taken as indicating a significant effect. Significance levels are reported as * ($p \leq .05$), ** ($p \leq .01$), and *** ($p \leq .001$). For all GLMM and LMM analyses reported, residual plots were visually inspected and revealed no obvious deviations from homoscedasticity or normality.

Fixations were categorised by interest area and GCA fixation-proportion analyses conducted in the *lmer* function [74, 75]. The nested repeated-measures design allowed for fixation proportions to the unrelated distractors to be averaged across distractor images. A third order (cubic) orthogonal polynomial was used to model the time course of fixation proportions on competitor type, and fixed effects of condition were included on all time terms. The baseline consisted of the no-competitor condition, and parameters were estimated for the two other conditions (cross-variety and same-variety competitor). Random effects of participants were included on all time terms. Estimates ($\beta$), standard errors (SE), and t-statistics are reported for all GCAs. The significance of effects was determined by assessing whether the associated t-statistics had absolute values of $\geq 2$ [76, 77].

## Results

### Comprehension task

**Response accuracy and latency.** Data trimming was performed prior to accuracy and RT analyses. Following a conservative approach to outlier elimination, responses above and below 3 SDs from the mean (lower bound = 976 ms, upper bound = 3290 ms) were considered outliers and excluded (additional 1.0% of data points removed). Accuracy was high across groups and conditions: The lowest group score was measured for the monovarietal interpreters in the

**Table 2. Mean comprehension-task RTs *(SD)* per group and condition.** Between-condition differences were not significant; non-interpreters performed significantly faster than interpreters.

| | Mean RTs *(SD)* per group and condition in ms | | |
|---|---|---|---|
| Group | Same-variety competitor | Cross-variety competitor | No competitor |
| Bivarietal interpreters | 2141 *(334)* | 2119 *(304)* | 2086 *(310)* |
| Monovarietal interpreters | 2192 *(379)* | 2200 *(374)* | 2177 *(369)* |
| Bivarietal non-interpreters | 2096 *(370)* | 2137 *(370)* | 2063 *(341)* |
| Monovarietal non-interpreters | 2065 *(360)* | 2109 *(345)* | 2086 *(347)* |

No significant interaction between interpreter and language-variety status was found ($\beta$ = - 0.037, SE = 0.045, t = -0.827), and RTs were neither affected by interpreter status ($\beta$ = -0.028, SE = 0.022, t = -1.275) nor by bivarietalism status ($\beta$ = 0.013, SE = 0.023, t = 0.579), nor competitor presence (No competitor vs. Swiss German competitor: $\beta$ = 0.016, SE = 0.018, t = 0.930; No competitor vs. Standard German competitor: $\beta$ = 0.002, SE = 0.018, t = 0.120); results reported are output from the reduced model providing a better fit according to the likelihood-ratio test performed: $\chi^2(1)$ = 0.679, p = .410.

same-variety competitor condition and was of 98.5%, (*SD* = 12.2) and all groups achieved 100% accuracy (*SD* = 0) in the baseline condition. No group effect was found: There was no significant interaction between bivarietalism and interpreter status ($\beta$ = 0.126, SE = 1.135, Z = 0.111, *p* = .911) and the reduced (interaction-free) model revealed no effect of either interpreter or language-variety status (interpreter status: $\beta$ = 0.621, SE = 0.568, Z = 1.094, *p* = .274; language-variety status: $\beta$ = -0.136, SE = 0.545, Z = -0.250, *p* = .802). No competitor effect on accuracy was found either (No competitor vs. L1' competitor: $\beta$ = -20.937, SE = 67055.476, Z = 0.000, *p* = 1; No competitor vs. L1 competitor: $\beta$ = -22.465, SE = 67055.476, Z = 0.000, *p* = 1).

Response latency (reaction time: RT) was measured to investigate a potential resource conflict as a function of task-completion speed. Trial completion took a minimum of 2063 ms (the fastest mean RT came from the bivarietal non-interpreter group on no-competitor trials) and a maximum of 2200 ms (the slowest mean RT came from the monovarietal interpreter group on cross-variety competitor trials). RT means per group are reported in Table 2.

**Fixation data.** The fixation-proportion data were analysed after excluding the 0.4% of incorrect trials to examine resource conflict as expressed in terms of co-activation of the language variety not used for the experimental task. Visual inspection of the competitor plots revealed a possible phonological competitor effect between 400 ms and 1000 ms after target-word onset, with diverging visual attention to the phonological competitors of the two varieties plotted in light grey (Swiss German competitor) and dark grey (Standard German competitor) compared to the distractors in that time frame plotted in black. The fixation-time course observed–a divergence of visual attention starting at around 400 ms after target-word onset–although later as found in other studies [78], is consistent with the theoretical framework of 200 ms estimated for lexical access [79] and the about 200 ms necessary to plan an eye-movement [80] and to fixate competitor and target [81]. Fig 3 shows the time course of fixation proportions on competitors compared to the baseline condition at 10 ms intervals (time bin size). The target-word onset in the auditory stimulus is plotted at 0. As can be gathered from Fig 3, all participants processed the same-variety competitors as competitors to the target, and while the monovarietal participants treated the cross-variety competitors like the other unrelated distractors, both bivarietal groups time courses indicate phonological cross-variety competition. Visually, the processing of both competitor types looks identical for the non-interpreter group, while a competitor-type distinction is visible for the interpreters, the same-variety competitors seemingly attracting more visual attention than the cross-variety competitors.

Significant effects of competitor type for the same-variety competitor condition were found for all participant groups, the two bivarietal groups also showed competitor effects for the Swiss German competitor condition.

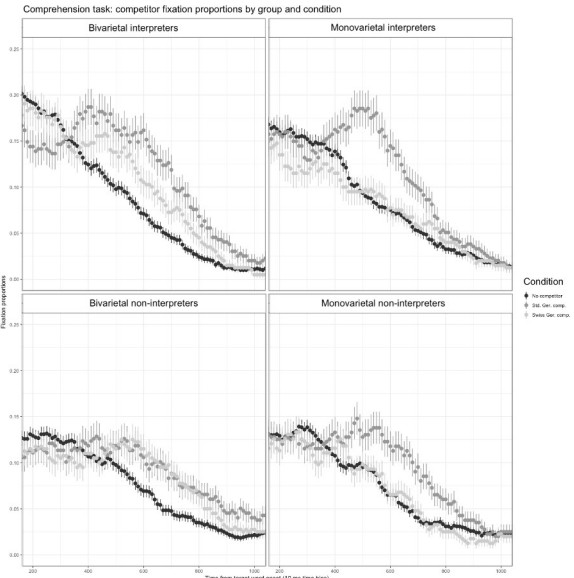

**Fig 3. Comprehension task: Competitor-type effect according to condition and split by participant group.**

The bivarietal interpreters showed an effect in both the same-variety competitor condition ($\beta$ = 0.052, SE = 0.009, t = 5.498) and the cross-variety competitor condition ($\beta$ = 0.025, SE = 0.009, t = 2.643) on the intercept term relative to the baseline, indicating that the gaze rested longer on competitors of both types compared to distractors overall. To verify whether the visual difference between the competitor types was significant, the baseline was relevelled to the Swiss German competitor condition, revealing that the proportion of fixations on the Standard German competitor was significantly higher than on the Swiss German competitor for that group ($\beta$ = 0.027, SE = 0.010, t = 2.737).

A model including task order, age and onset of fluency in German as fixed effects was used for an initial analysis to verify whether these variables affected the results. None of the additional variables had a significant effect, and the significance of the other effects remained unchanged in the larger model (see S3 Appendix for an overview of the full and reduced analysis models as well as S4 Appendix for the model outputs).

As hypothesised and visible in Fig 3, the monovarietal interpreters showed a significant competitor effect only for the same-variety competitor condition ($\beta$ = 0.041, SE = 0.010, t = 4.206). The cross-variety competitor condition was processed like the distractor baseline ($\beta$ = 0.002, SE = 0.010, t = 0.208). Cross-variety competitors, therefore, attracted the same amount of overt visual attention as the distractors.

The bivarietal non-interpreters' gaze behaviour matched the bivarietal interpreters', showing an effect in both the same-variety competitor condition ($\beta$ = 0.029, SE = 0.011, t = 2.659) and the cross-variety competitor condition ($\beta$ = 0.024, SE = 0.011, t = 2.163) indicating more looks to competitor objects of both types relative to unrelated objects. However, unlike the bivarietal interpreters, the bivarietal non-interpreters showed no difference in fixation proportions between the same-variety and the cross-variety competitors ($\beta$ = 0.006, SE = 0.013, t = 0.417; see Fig 3). An analysis of the active use (i.e., producing spoken utterances) of the two varieties in the two bivarietal groups as indicated in the post-hoc questionnaire revealed that the interpreters used Standard German significantly more often than non-interpreters do (see Table 3; p < 0.5, Cohen's D = 2.8), which could explain the higher degree of activation of the Standard variety in bivarietal interpreter participants.

**Table 3. Between-group comparison use of Standard German vs. non-standard varieties.** Δ indicates the differences in the percentage of Standard German use between the groups, * indicates significance.

| | Mean (*SD*) frequency of use of Standard German in % | | |
|---|---|---|---|
| Bivarietal groups | Non-bivarietal groups | | Δ |
| 36.3 (*24.5*) | 91.7 (*15.4*) | | 55.4* |
| | Mean (*SD*) frequency of use of Standard German in % | | |
| Bivarietal interpreters | Non-bivarietal interpreters | | Δ |
| 44.4 (*26.1*) | 28.1 (*17.9*) | | 16.3* |

Both monovarietal groups only showed a gaze-behaviour effect for the same-variety competitor condition ($\beta$ = 0.032, SE = 0.010, t = 3.304), not for the cross-variety competitor condition ($\beta$ < 0.001, SE = 0.010, t = 0.023, see Fig 3). The within-group competitor comparisons hence reveal that a same-variety competitor effect was present in all groups. Between-group analyses of competitor activation by competitor-type revealed no changes in same-variety competitor activation as an effect of group profile. The activation of the same-variety competitor in the two bivarietal groups was comparable ($\beta$ = 0.021, SE = 0.017, t = 1.243). Therefore, although bivarietal interpreters distinguished between the Swiss-German and the Standard German competitor types, while the bivarietal non-interpreters did not, the between-group difference in terms of fixation proportion of the Standard German competitor was not significant. The same applied to the two monovarietal groups ($\beta$ = 0.014, SE = 0.017, t = 0.838). Therefore, no significant effect of SI experience on fixation proportions was found. The activation of the Standard German competitor in the two interpreter groups was also comparable ($\beta$ = -0.006, SE = 0.014, t = -0.403). Thus, no significant effect of bivarietalism on fixation proportions on the Standard German competitor was found either. The activation of the Swiss German competitor in the bivarietal non-interpreters compared to the bivarietal interpreters was also not significantly different ($\beta$ = -0.001, SE = 0.015, t = -0.054), however, the cubic time term reached significance ($\beta$ = 0.060, SE = 0.029, t = 2.054), indicating a shallower curvature, which we interpreted as a tendency to a lower proportion of fixations to the cross-variety competitor in the bivarietal interpreter group. However, although the bivarietal interpreters, unlike their non-interpreter counterparts, distinguished between the two competitor types, the between-group difference in fixation proportions on the Swiss German competitor was not significant.

Visual inspection of the competitor plots in Fig 3 raised the question of whether fixation levels were overall higher in interpreters than in non-interpreters. However, subtracting the no-competitor baseline values from the two competitor conditions and comparing the fixation proportions between groups revealed no significant between-group difference (bivarietal interpreters vs. monovarietal interpreters: $\beta$ = 0.005, SE = 0.013, t = 0.381; vs. bivarietal non-interpreters: $\beta$ = 0.002, SE = 0.013, t = 0.137; vs. monovarietal non-interpreters: $\beta$ = 0.001, SE = 0.013, t = -0.035). This was confirmed by relevelling the fixed effect of group to the groups other than the bivarietal-interpreter group's as the baseline.

## Production task

**Response accuracy and latency.** Accuracy scores were gathered manually. Accuracy was again high across tasks, groups and conditions (from 92.6%, *SD* = 26.7, which was the lowest score, measured for the bivarietal interpreters in the same-variety competitor condition, to 97.9%, *SD* = 14.5, also for the bivarietal interpreters, but this time in the baseline condition; and from 81.3%, *SD* = 40.0 for the bivarietal non-interpreters in the cross-variety condition to 96.3% for the monovarietal non-interpreter group, again in the baseline condition). As for the

comprehension task, no group effect on accuracy was found in either task: Language-variety status did not significantly alter performance in any of the conditions (interpreters: language-variety status: $\beta$ = -0.120, SE = 0.322, Z = -0.373, p = .709; No competitor vs. Swiss German competitor: $\beta$ = -1.183, SE = 0.810, Z = -1.461, p = .144; No competitor vs. Standard German competitor: $\beta$ = -0.341, SE = 0.845, Z = -0.404, p = .686). However, differences were found for the non-interpreters: language-variety status significantly affected accuracy, i.e., the bivarietal group's accuracy rates were significantly lower ($\beta$ = 0.846, SE = 0.330, Z = 2.565, p = .010). Accuracy on trials with Swiss German competitors was also significantly lower compared to the baseline ($\beta$ = -2.466, SE = 1.163, Z = -2.121, p = .034). In contrast, the Standard German competitor did not affect accuracy (No competitor vs. L1 competitor: $\beta$ = -0.859, SE = 1.165, Z = -0.737, p = .461). The lower accuracy scores for the bivarietal group are explained by a more frequent use of synonyms for the target words (trials that did not contain semantic errors, but where a synonym that could not have triggered phonological competition were excluded from the analysis).

RT measures were extracted using a Praat [82] script and analysed analogously to the comprehension task to explore whether a potential conflict of resource use affected task-completion speed. The experimenter manually checked the Praat output when verifying the semantic output to determine accuracy scores. The minimum mean RT recorded for the interpreting variety of the PT was of 1938 ms ($SD$ = 691; bivarietal interpreters on Standard German competitor trials), the maximum RT was of 2064 ms ($SD$ = 582; monovarietal interpreter group on Swiss German competitor trials), for the target-word translation task, the monovarietal group averaged the fastest replies in the Standard German competitor condition (1485 ms, $SD$ = 417) and the slowest replies came from the bivarietal group on Swiss German competitor trials (1675 ms, $SD$ = 494). No significant interaction between language-variety status and condition was found for the interpreters (language-variety status x Swiss German competitor: $\beta$ = 0.006, SE = 0.025, t = 0.228; language-variety status x Standard German competitor: $\beta$ = 0.012, SE = 0.025, t = 0.490), and neither language-variety status nor condition significantly affected RTs (language-variety status: $\beta$ = 0.018, SE = 0.065, t = 0.280; No competitor vs. Swiss German competitor: $\beta$ = 0.043, SE = 0.032, t = 1.350; No competitor vs. Standard German competitor: $\beta$ = -0.014, SE = 0.032, t = -0.440; results reported are again the reduced-model output without interaction term, as the result of the likelihood-ratio test indicated a better fit: $\chi2(1)$ = 0.679, $p$ = .410.). The same pattern was observed for the performance of the non-interpreter groups on their version of the production task: No significant interaction between language-variety status and condition could be established (language-variety status x Swiss German competitor: $\beta$ = -0.005, SE = 0.026, t = -0.187; language-variety status x Standard German competitor: $\beta$ = -0.043, SE = 0.025, t = -1.705). The output of the reduced model used for analysis after the non-significant result of the likelihood-ratio test ($\chi^2(2)$ = 3.405, p = .182) indicated no effect of language-variety status or of the presence of either competitor type on RTs (language-variety status: $\beta$ = -0.064, SE = 0.039, t = -1.639; No competitor vs. Swiss German competitor: $\beta$ = 0.056, SE = 0.049, t = 1.156; No competitor vs. Standard German competitor: $\beta$ = -0.021, SE = 0.048, t = -0.433).

**Fixation data.** Given that non-interpreters did not have any experience with SI, only interpreter participants were instructed to simultaneously interpret the English sentences into German. Non-interpreter participants were asked to only name the sentence-final target object in German. Given this important task difference, the two resulting data sets were analysed separately. The analysis approach described for the comprehension task was also applied to the production-task data, resulting in the removal of 9.1% of the data for the interpreting variant and 12.2% for the target-word translation variant of the task due to inaccurate responses. Outlier removal (+/-3SDs from the overall mean, the lower bound was set at 400 ms) resulted in

the exclusion of 13% of the RT data points for the interpreting variant and of 5.0% of RT data points for the target-word translation variant of the production task.

Fixations were again analysed according to interest areas and distractors were averaged. As there is evidence for an effect of phonological competition of translation equivalents in terms of visual attention [83], three time windows were inspected for phonological competition during production. The first time window started 600 ms post target-word onset, under the assumption that L2 lexical access and ensuing L1 retrieval take 200 ms each and by adding 200 ms for planning and executing an eye movement based on the retrieval of the L1 word form of the target. To verify whether speech planning might be the driver of eye movements [67–72] and thus possibly a source for phonological competition in the target language, the second time window was set for shortly prior to the onset of the target-word production in the participants' L1 output. Should however the processing of self-generated speech [59] rather than speech planning, or the retrieval of the L1 word form from the mental lexicon, lead to phonological competition, then effects would be visible only once the L1 target is articulated and reprocessed. The third window of analysis thus started roughly 200 ms after the onset of production of the target-word translation equivalent.

Unlike for the comprehension-task data, however, and as illustrated in Fig 4(A) for the interpreting-task variant and Fig 4(B) for the final-word translation-task variant, no phonological competitor effect was discernible from visual inspection of the plots:

While the factors *task order* and *age of L2 onset* had no effect on the results, *sentence rating* did ($\beta$ = -0.003, SE = 0.001, t = -3.566) and was therefore added to the analysis models. *Language-variety status* was the factor that set the two groups apart for both the interpreting and the target-naming variants of the production task. However, as Table 4 shows, there was no significant main effect of language-variety status on co-activation in either task in any of the time windows:

In window 1, a significant difference in terms of attracting visual attention was noted for the group of monovarietal interpreters in their task variant, however against expectations, the Standard German competitor was attended to significantly less than the distractors. Both non-interpreter groups showed no same-variety competitor effect during that same time window, but paid significantly more attention to the Swiss German competitors, which was particularly unexpected for the monovarietal group. With the exception of bivarietal non-interpreters attending significantly more to the Standard German competitors during time window 2, neither competitor type attracted significantly more visual attention than the distractors in any of the other time spans analysed, as the results reported in Table 5 show:

## Discussion

This study set out to investigate whether phonological language-variety co-activation can be observed analogously to bilingual co-activation in comprehension and production, and whether the type of task performed and professional SI experience influences such co-activation.

The fixation-data analysis revealed that all participant profiles experienced same-variety phonological competition during comprehension and that Swiss-German co-activation in the bivarietal groups followed the predicted pattern–to our knowledge a novel finding. The hypothesis that language-variety co-activation occurs and follows the same activation patterns as the co-activation of a separate language system, was thus corroborated for comprehension. SI experience, however, did not condition the co-activation of either language variety, even though for interpreters' output to meet their audience's needs, they are not only required to render the semantic content of the message; they also select the appropriate output language

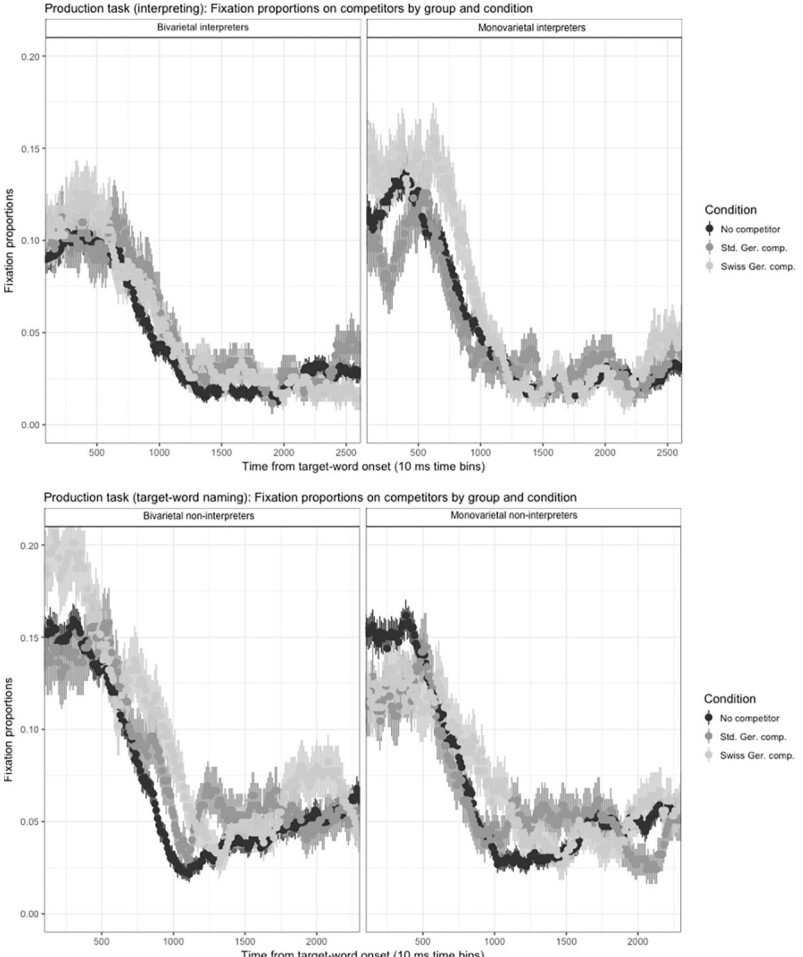

**Fig 4. A**. Production task : Competitor-type effect in the interpreting variant by group. **B**. Competitor-type effect in the sentence-final word translation variant by group.

variety and habitually adapt their speech register to match the original [84, 85], which led us to assume a higher sensitivity to register differences in interpreters than non-interpreters at the outset. It thus appears that, contrary to initial assumptions, SI training and professional practice do not affect co-activation patterns for comprehension. However, an interesting difference between the gaze patterns of bivarietal interpreters and non-interpreters was noted: Both groups experienced competition from both competitors, but bivarietal non-interpreters fixated both competitor types to the same extent, while bivarietal interpreters fixated the same-variety competitor significantly more than the cross-variety competitor. As the interpreter-status effect was not significant, this between-group difference could not be attributed to a change in activation levels as a result of SI experience. Furthermore, the competitor effects reflected in

**Table 4. Analysis of the effect of bivarietalism in the production tasks for the analyzed time windows.**

| Time window | Task variant: interpreting | Task variant: target-word translation |
|---|---|---|
| 1 | $\beta < 0.001$, SE = 0.010, t = 0.008 | $\beta = -0.011$, SE = 0.010, t = -1.034 |
| 2 | $\beta = 0.007$, SE = 0.006, t = 1.199 | $\beta = 0.003$, SE = 0.009, t = 0.294 |
| 3 | $\beta = -0.001$, SE = 0.011, t = -0.049 | $\beta < 0.001$, SE = 0.011, t = -0.022 |

**Table 5. Analysis of the effect of competitor type in the production tasks for the analyze time windows.** Analysis of phonological competition as measured in fixation proportions during the production task.

| Window | Competitor type | Task variant: interpreting | | Task variant: target word translation | |
|---|---|---|---|---|---|
| | | Bivarietals | Monovarietals | Bivarietals | Monovarietals |
| 1 | Standard German | $\beta$ = -0.001, SE = 0.012, t = -0.069 | $\beta$ = -0.040, SE = 0.010, t = -4.111 | $\beta$ = 0.020, SE = 0.012, t = 1.778 | $\beta$ = 0.002, SE = 0.010, t = 0.258 |
| | Swiss German | $\beta$ = -0.007, SE = 0.012, t = -0.593 | $\beta$ = 0.009, SE = 0.010, t = 0.899 | $\beta$ = 0.038, SE = 0.012, t = 3.263 | $\beta$ = 0.024, SE = 0.010, t = 2.417 |
| 2 | Standard German | $\beta$ = 0.010, SE = 0.007, t = 1.498 | $\beta$ = 0.007, SE = 0.008, t = 0.916 | $\beta$ = 0.029, SE = 0.013, t = 2.232 | $\beta$ = 0.023, SE = 0.013, t = 1.807 |
| | Swiss German | $\beta$ = 0.005, SE = 0.007, t = 0.765 | $\beta$ = 0.003, SE = 0.008, t = 0.334 | $\beta$ = 0.009, SE = 0.013, t = 0.742 | $\beta$ = 0.008, SE = 0.013, t = 0.652 |
| 3 | Standard German | $\beta$ = 0.004, SE = 0.007, t = 0.545 | $\beta$ = 0.004, SE = 0.009, t = 0.447 | $\beta$ = 0.001, SE = 0.014, t = 0.072 | $\beta$ = -0.006, SE = 0.012, t = -0.523 |
| | Swiss German | $\beta$ = -0.007, SE = 0.007, t = -1.067 | $\beta$ = 0.009, SE = 0.009, t = 0.969 | $\beta$ = 0.001, SE = 0.014, t = 0.078 | $\beta$ = -0.022, SE = 0.012, t = -1.839 |

the eye-tracking measures were absent in the accuracy and RT data. Neither interpreter or language-variety status, nor competitor presence thus affected performance accuracy or speed. The data bore out no significant effect of bivarietalism status either, however, the bivarietal interpreter group tended to allocate less visual attention to the cross-variety competitor as suggested by the significant cubic time term (see Results) indicating a shallower curvature. Based on the assumption that lexical activation is not a process that can be captured in a linear manner, but that its time course (and according to our hypothesis therefore also the time course of language and language variety co-activation) would follow a sinusoidal rather than a linear or sigmoidal curvature, the cubic term parameter was taken as an indicator of the curvature of lexical activation as expressed in visual attention indicating an increase and subsequent decrease of visual attention after the disambiguation point during comprehension, rather than a continuous shift [86]. Although there are other valid approaches to analysing Visual World fixation data besides GCA [see e.g., 81], we opted for this analytical approach in an attempt to capture the up-and-down movement expected in terms of fixation proportions to the phonological competitor in a time window selected according to theoretical rationale ahead of the analysis and on the basis of the time estimated necessary according to evidence from the literature for lexical access and planning and executing the eye movement, while also jointly accounting for by-subject and by-item variability. As a the two bivarietal groups presented a significant difference regarding the use of the two varieties, it is possible that frequency of use and degree of activation are linked, suggesting that actively using a language (as in actively using it to speak and as opposed to passive exposure) may increase activation levels and therefore improve lexical access. Not least for simultaneous interpreters who rely on quick lexical access to properly do their job, empirically corroborating a strategy that is commonly taught based on a hunch is of considerable relevance.

The data indicating no SI-practise related difference regarding co-activation during comprehension, while still revealing a cross-variety competitor activation difference between the two bi-varietal groups seems inconsistent. However, this difference (that did not reach significance in between-group comparisons) can be interpreted as bivarietal interpreters being less distracted by the cross-variety competitor, it could also be construed as them being initially more distracted by same-variety competition, but then being able to resolve the competition efficiently–more efficiently than the bivarietal non-interpreters, whose fixation proportions run along a less steep curvature and neither reach the matching interpreter groups maximum height, nor the ensuing lowest point. This could point to repeated exposure to co-activation (which arguably is the case during simultaneous interpreting) increasing co-activation contrary to our initial hypothesis, but that having to routinely resolve co-activation induced phonological competition makes that process swifter. Unfortunately, our data do not allow us to corroborate this claim.

As also helpfully pointed out by a reviewer, however these findings are supported by previously reported evidence [87], showing that recurring co-activation results in competition becoming more expeditiously resolved, and demonstrating in TRACE simulations [see 88 for details on the TRACE model] that an increase in competition comes with an increase in competitor inhibition.

In conclusion, although our data do not indicate that the extreme form of multilingual–and multivarietal–processing that is part of SI leads to a distinct processing advantage (neither significantly lower co-activation nor significantly more efficient phonological competition resolution), the difference found between the two bidialectal groups and their unequal processing of the same-variety compared to the cross-variety competitor could be a promising angle to further investigate the role of competition resolution by simultaneous interpreters.

Contrary to comprehension, the gaze patterns measured during production did not indicate systematic overt attention directed at either type of phonological competitor in either task variant and participant profile, nor for any of the three time windows analysed: window 1: 600–1200 ms–phonological competition to the translation equivalent of the auditory target input upon target-input perception and translation-equivalent retrieval; window 2: 1500–2100 ms–phonological competition during speech planning prior to articulation; time window 3, 2000–2600 ms–phonological competition via participants' own perception system due to their reprocessing of their own target-word production. In other words, not only was there no evidence for consistent co-activation of the non-target variety of Swiss German for the bivarietal participants–only in time window 1 did the bivarietal non-interpreter group pay significantly more visual attention to the cross-variety competitor compared to the distractors, however, the monovarietal non-interpreters, for whom the cross-variety phonological competitors were unrelated distractors, also showed a significant effect. At the same time, with two exceptions, no significant co-activation of same-variety cohort competitors was found for any of the groups: An unexpected negative significant same-variety competitor effect was measured for monovarietal interpreters in time window 1 (i.e., the Standard-German competitor was attended to significantly less) and bivarietal non-interpreters also showed a significant effect of the same-variety competitor in time window 2, however, this was the only significant same-variety competitor effect measured for that group and not part of a larger pattern. As the frequency of the target, competitor and distractor names was controlled for German and English, and as unwanted phonological competition from the French and English labels was excluded during stimulus design, the cause of the observed effects could not be assigned to object-label features. In line with the results of the fixation-data analysis, *language-variety status* or *competitor presence* did not affect response accuracy or latency. The effect found during the speech perception phase in window 1 for the two non-interpreter groups is therefore not likely to be linked to phonological competition. We cannot, however, exclude other individual differences potentially entailing variations in co-activation [see 89]. During the third stage of processing (window 3) that included the articulation and re-processing of the target word for which phonological competitors were present for all participants in the same-variety competitor condition, no indication of gaze diversion and thus of measurable phonological competition was found. Additional research would be needed to investigate whether the speech-planning phase may be more sensitive to phonological competition during production, if such an effect can be established at all, or to investigate a potential qualitative difference in terms of retrieval of the phonological form during comprehension and production. Response accuracy was not affected by the presence of phonological competitors, however, bivarietal non-interpreters' answers were significantly less accurate than their monovarietal counterparts'. Error analyses revealed that bivarietal participants used synonyms almost twice as often as monovarietal participants. Their answers were thus not wrong semantically speaking, which attenuates the

significant between-group difference found, but the diverging word forms used had to be excluded from the analysis as they could not trigger phonological competition. Even though the monovarietal participants tended to respond faster, no significant between-group RT difference was found. Competitor type also had no significant effect on RTs. Based on the measures gathered, no indication of phonological competition on either variant of the production task was found.

It may be important to consider that the comprehension task required the processing of an audio input followed by a motor response (selecting the image corresponding to the audio input, moving the mouse cursor and clicking on that image), while the production task required the processing of an L2 audio input, the translation of the whole input (interpreter variant) or the sentence-final target word (non-interpreter variant) into the participants' L1 as well as the articulation of the translation. This difference in the nature of the experimental tasks–the comprehension task requiring two motor responses (eye and hand movements) vs. production requiring a motor and a verbal response (eye movements and articulation of the translated sentence or target-object name), but also in comparison with the production studies, which previously found evidence for co-activation during production, but that did not comprise a language-transfer element [59, 67–69, 83]. As one reviewer helpfully suggested, it is possible that the comprehension task allowed for an easier association of responses of the same type than the second task that associated distinct response types. We indeed cannot exclude that co-activation effects lie hidden behind a more cumbersome response association for the production task in an overtly multilingual setting.

The absence of the expected competitor effects in both varieties of the production task data is of course far from an indication of an actual absence of such an effect. As indicated, we cannot exclude that the remaining complexity of the production tasks designed for this study, for which we attempted to make a complex task presenting a great degree of variability lab and measurement friendly, may have obscured effects or prevented them from manifesting due to a processing bottleneck or the exhaustion of available cognitive resources. It is also possible that the response-modality complexity in particular (going from processing an audio input to manually selecting the corresponding image on a screen, which previous studies have demonstrated to be a robust design to investigate the effect we were interested in to processing an L2 audio input and having to articulate an L1 response for which a cross- or same-variety phonological competitor was presented) may not have allowed to pinpoint phonological competition, again by overcharging the resources available, or by pushing for target disambiguation before competition could measurably come into play. If that were the case, this may also call into question the sensitivity of measures used, which may then not have been sufficiently adapted or adequate to pick up effects. We furthermore have to concede that we cannot exclude that ceiling effects regarding the resource allocation as we were able to measure and analyse it may have obscured co-activation effects present in the process.

In summary, while our data bears no evidence of co-activation during production, a distinct same-variety effect was measured during comprehension for all participants as well as a distinct cross-variety effect for bivarietal participants. Contrary to expectations, the SI variable– the ability to routinely perform a multilingual task that is thought to require an extraordinary cognitive control effort–did not affect processing in terms of activation, response accuracy or response latency. As the two production tasks were different and differences in underlying processes cannot be excluded, direct comparisons are futile. However, the two groups who performed the same task differed regarding their bivarietalism status and still showed no competitor effect at all. While the absence of a measurable effect in this set-up is insufficient evidence to claim an absence of co-activation, the experiment results are still of considerable interest from a language-processing perspective.

First, our data provide evidence for co-activation of task-irrelevant non-standard varieties that follows the same pattern as co-activation of typologically very different language pairs [e.g., 1, 2 looked at English and Russian; 3 at English and French, and 6 at English and Mandarin Chinese]. On the one hand, a more nuanced view and possibly selection of participants may well be required for multilingualism to account for bivarietalism. On the other hand, certain questions could potentially also be addressed using bivarietal or multi-varietal participants without their findings losing validity for multilinguals.

Second, while co-activation seems to be a fine-grained yet robust phenomenon during comprehension, measuring potential co-activation during a production task involving language transfer appears more challenging.

Where an effect was found, extreme multilingual processing does not seem to affect it *en bloc*–neither by strengthening it to make lexical entries more accessible, nor by weakening it to lower the demands on cognitive control. This goes against the long-held hypothesis that the cognitive task demands of SI lead to domain-general changes. However, as discussed above, further investigation of competition resolution efficiency may add nuance to this rather black-and-white conclusion. A more delicate approach to the question would also be in line with other cognitive capacities long viewed as simply superior in simultaneous interpreters. Working-memory capacity, for example, has been widely investigated with a view to pinpoint possible effects of SI expertise on the memory component of processing [90–92; see 93 for evidence on potential effects of bilingualism on working memory]. The data presented here do not suggest a correlation, which is in line with evidence suggesting a more tenuous link between domain-general and language-specific cognitive control [94–98, cf. 99, 100] than previously assumed [101–103]. Even assuming that simultaneous interpreters are cognitive control experts [17–19], investigating potential control-process changes does not seem to reveal systematic significant effects in behavioural measures [104]. Brain-activity measures, however, appear to be more sensitive to such changes [12, 105–109].

Third, while no evidence was found indicating that expertise in SI alters language-variety co-activation patterns, the frequency of active language-variety use appears to have a bearing on activation, as indicated by the higher activation levels measured for Standard German in the interpreter compared to the non-interpreter bivarietal speakers and the significant difference in active Standard German use between the two groups. However, frequency of language use and language proficiency are hard to untangle [110]. Language proficiency is complex and the extent to which it may alter processing has not been established [21, 22, 106]. Additional measures on language-variety background not available for the participants of this study would be necessary for further analyses in this respect. An additional limitation to be considered when interpreting the results presented is that while great care was taken to recruit participants with relevant language and skill profiles, the specific demands regarding bivarietalism and the intersection with SI skills severely restricted the pool of potential participants. While the rule of thumb for analyses in linear mixed models [111] was adhered to for the main comparisons, this was not possible for more fine-grained analyses.

Studying language processing in simultaneous interpreters has been put forward as an entry point to unveiling otherwise elusive processes owing to the extreme task demands of SI. In other words, by studying verbal and non-verbal processing in simultaneous interpreters, task-induced processing differences could become more evident. However, we cannot exclude that integrating the task and the corresponding control mechanisms with increasing expertise may render them invisible to our measures. However, from the implications of the results discussed, we surmise that even the extreme processing feat of SI does not give interpreters a general advantage in avoiding co-activation of task-irrelevant language varieties–and presumably

languages. To our knowledge, however co-activation of context-irrelevant languages in interpreters has not been investigated to date.

From the findings discussed above we conclude that the factor with the strongest impact on co-activation appears to be relative frequency of active language use (the more speakers actively produce utterances in a specific language or variety, the stronger the activation level of that language or variety)–an insight that seems to apply to languages and to varieties alike. Further investigation is necessary to establish whether more effective competition resolution in simultaneous interpreters could come as a consequence of stronger co-activation, for which it would be interesting to also explore the role of language proficiency in processing and the strength of networks within which languages or language subsets are organised.

The results presented here provide evidence for co-activation on a fairly fine-grained level of linguistic systems and sub-systems during comprehension. To date, monolingualism has generally been viewed as monolithic: a fixed parameter rather than a variable. As such, it has then been contrasted with multilingualism, i.e., the mastery of more than one language system, although the view of multilingualism as a continuous variable is gaining ground. As the present findings point to co-activation extending to language varieties, they not only suggest that the study of bivarietalism could be of relevance to shed further light on the ongoing discussion regarding multilingualism, they also allow us argue that monolingualism is a much less monolithic linguistic experience than long assumed, and that the net for capturing the full variety of multilingual experience must be cast wider.

## Supporting information

**S1 Appendix. Stimulus sets with characteristics.**
(PDF)

**S2 Appendix. Sentence lists.**
(PDF)

**S3 Appendix. Analysis models.**
(PDF)

**S4 Appendix. GCA model scripts and outputs.**
(PDF)

**S5 Appendix.**
(PDF)

**S1 File. Minimal data set.**
(ZIP)

## Acknowledgments

The authors are most grateful to Dr. Rhona Amos, Research and Teaching Fellow, University of Geneva, and Dongpeng Pan, Doctoral Candidate, University of Geneva, for the fruitful exchanges and for sharing their insights on the intricacies of GCAs and the interpretation of the model outputs.

## Author Contributions

**Conceptualization:** Laura Keller.

**Data curation:** Laura Keller.

**Formal analysis:** Laura Keller, Malte C. Viebahn.

**Investigation:** Laura Keller.

**Methodology:** Laura Keller, Alexis Hervais-Adelman.

**Project administration:** Laura Keller.

**Supervision:** Alexis Hervais-Adelman, Kilian G. Seeber.

**Visualization:** Laura Keller, Malte C. Viebahn.

**Writing – original draft:** Laura Keller, Malte C. Viebahn, Alexis Hervais-Adelman, Kilian G. Seeber.

**Writing – review & editing:** Laura Keller.

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
