## [Decision Letter · Decision Letter 0]

30 Jan 2023

PONE-D-22-33159Unpacking the multilingualism continuum: An investigation of language variety co-activation in the context of simultaneous interpretingPLOS ONE

Dear Dr. Keller,

Thank you for submitting your manuscript to PLOS ONE. After careful consideration, we feel that it has merit but does not fully meet PLOS ONE’s publication criteria as it currently stands. Therefore, we invite you to submit a revised version of the manuscript that addresses the points raised during the review process.

We look forward to receiving your revised manuscript.

Kind regards,

Nicola Molinaro, Ph.D.

Academic Editor

PLOS ONE

Journal Requirements:

Additional Editor Comments:

The Reviewers found a number of positive aspects of your study, but, at the same time, they raise a number of issues that should be considered by the authors.

Reviewers' comments:

Reviewer's Responses to Questions

**Comments to the Author**

1. Is the manuscript technically sound, and do the data support the conclusions?

Reviewer #1: Yes

Reviewer #2: Yes

2. Has the statistical analysis been performed appropriately and rigorously? 

Reviewer #1: Yes

Reviewer #2: Yes

3. Have the authors made all data underlying the findings in their manuscript fully available?

Reviewer #1: Yes

Reviewer #2: Yes

4. Is the manuscript presented in an intelligible fashion and written in standard English?

Reviewer #1: Yes

Reviewer #2: Yes

5. Review Comments to the Author

Reviewer #1: In all, this is a sound piece of work. It is innovative in that it demonstrates the relevance of language variation in activation patterns. It is methodologically sound in that it exemplarily controls the experimental conditions.

Here are some suggestions to improve the paper in some respects.

1. The title very ambitiously touts simultaneous interpreting, but none of the tasks, except one, actually involve any kind of simultaneous interpreting. I’d go for something like “simultaneous interpreters”.

2. The relevance of simultaneous interpreting (SI) or relevance thereof in co-activation could be better stated, especially because the paper proposes a specific hypothesis that is difficult to connect with the short description of SI. The main idea seems to be that co-activation is reduced to offset higher cognitive demands (comprehension higher vs. production lower, end of p. 7), but that experience in SI may again enhance co-activation (non-interpreter lower vs. interpreters higher, top of p. 8). There is some doubt in the mind of this reviewer because in lines 171-173, the authors write that “experience with SI may alter the degree of co-activation of task-irrelevant varieties due to high cognitive demands of the primary task.” The verb “alter” is not specific about the orientation of the trend and the logical conclusion readers now draw from the framing of the sentence is that co-activation would be lower in interpreters. Production is more cognitively demanding than comprehension, so activation is lower in production; ergo if SI is more cognitively demanding, activation is lower in interpreting. Or is the whole argument based on the idea that experience leads to automation, reducing cognitive demands on the interpreting task, leading to higher co-activation? If that’s the case, the missing link of automation should be included. Either way, the wording on top of p. 8 should be “altered” to make it less contradictory.

3. Interpreters’ awareness of register differences is hypothesised to account for a higher degree of co-activation of the standard German competitor. That seems fair, but it was not the focus of the research. It is therefore better to reserve the topic of register to the discussion, in stead of smuggling it into the introductory part on SI.

4. The discussion could also perhaps cover an important difference between the two tasks, namely that the first task was a combination of audio input and motor response (selecting and clicking), while the second task was a combination of audio input and articulatory response. Eye movement being a motor response, the first task perhaps more easily associates with another motor response than the second?

Minor issues:

L 112: “Unlike monolingual communication” : the authors just stated that co-activation is constant.

L 163: “language” > “languages”

Table 1, General, Age-mean (end of line): G3 should probably be G4

L 253-254: “without a strong regional affiliation”: how did the authors check that?

L 255 sq: the whole idea of disambiguation is unclear.

L 333 ‘to look’ at > ‘to look at’

Figure 3: impossible to read. The original picture that can be downloaded through the link is fine, but the one that comes with the paper is unclear.

L 482: if time-out was set at 3000 ms (L 298), how can the upper bound be 3290? Participants clicked on an image they no longer saw?

L 546 sq: it is misleading to write that there are no significant signals, as the Table shows that there are. In the discussion the data are correctly represented, but it should be done here too.

L 655 sq: is a bit of an exaggeration. Only bivarietal non-interpreters come close to t≥2.

Reviewer #2: Article summary: The article presents an experiment that examines phonological co-activation/competition within and across language varieties. Specifically, it asks whether co-activation similar to that observed between languages can also be observed between dialects and whether (and how) such co-activation may be affected by: a) task type (comprehension versus production), b) language use, and c) experience in simultaneous interpreting. To address these questions, the authors tested all German L1 speakers varying orthogonally in 1) their knowledge of a non-standard German dialect (Swiss German) and 2) in whether they were trained simultaneous interpreters. Phonological co-activation was assessed via two standard Visual World Paradigm tasks (one on comprehension and one on production). The results showed phonological co-activation within the same dialect, replicating previous results, but also expanded these results by showing a similar pattern for inter-dialectal co-activation. Experience in simultaneous interpreting seemed to have an effect in some analyses, as interpreters-only were more distracted by intra-dialectal competitors. However, for the production task the results did not show evidence for phonological co-activation of either type.

Review summary: This is a very nice study that touches on interesting questions expanding on previous work on cross-language co-activation. The literature review is thorough, the hypotheses and predictions are clearly spelled-out, and the experimental methods are laid down in a well-organized manner. Finally, the writing is clear and the narrative is easy to follow.

Although this work has many positive features, there are a few issues that need to be addressed before publication.

Major issues and suggestions:

1. The authors do a great job at reporting details of their analytical approach; however, there is some important information missing. The authors use growth curve analyses (GCA) and specifically a third-order (cubic) polynomial. My understanding is that this type of analysis produces results related to: an intercept, a linear, a quadratic, and a cubic term. However, from the manuscript it is not clear which parameter(s) the results correspond to (with the exception of one result specified as a cubic term effect in line 465). If I had to guess, I would say that the rest of the results correspond to the intercept, but in that case, it is not clear what the advantage is of using a dynamic analytical approach such as GCA instead of a traditional fixation probability measure. In any case, I am not suggesting that the authors change their analytical approach; I simply ask for some more clarity and transparency. For example, the authors could report the full models and their outputs. In addition, it would help to have some clear information as to the authors' interpretation of the corresponding parameter (i.e., the cognitive process reflected in that parameter).

2. It is not clear what the overall conclusion is regarding the effect of experience with simultaneous interpreting on inter-/intra-dialectal co-activation. On the one hand, the authors conclude that “contrary to initial assumptions, SI training and professional practice do not affect co-activation patterns for comprehension”, but later on they mention that “bivarietal non-interpreters fixated both competitor types to the same extent, while bivarietal interpreters fixated the same-variety competitor significantly more than the cross-variety competitor”. I understand that the data are a bit contradictory in that respect; however, I think that this is an important point that deserves to be fleshed out a bit more.

Related to this, I found it surprising that the authors do not come back to the significant effect on the cubic term (see lines 464-467). Looking at Figure 3 with this effect in mind, I can’t help but notice that bivarietal interpreters not only seem to be more distracted by same dialect competitors, but they also seem to do a much better job at resolving phonological competition: competitor looks for interpreters go up to .17-.20 and they manage to get them down to .0-.025 by the end of the trial, while for non-interpreters, competitor looks go up to only .125 and still they don’t manage to get them as low as interpreters. Interestingly, this pattern is very much in line with work reported by Kapnoula & McMurray (2016), in which we saw that a) more frequent co-activation leads to more efficient competition resolution and b) TRACE simulations showed that higher/stronger competition is accompanied by stronger inhibition of the competitor item. I am not sure how the authors could look a bit more closely into this pattern and compare competition resolution across participant groups using GCA, but if this pattern were to be backed up by stats, then one could say that SI *can* in fact give interpreters an advantage, not in terms of avoiding co-activation of task-irrelevant languages / language varieties, but in terms of resolving phonological competition more effectively.

3. Finding no evidence for co-activation in production should be interpreted with more caution, given that there are a few reasons why this could be the case:

a) First and foremost, in the production task, participants were not required to click on the target. This means that they could perform the entire task without ever having to look at the images on the screen. This is likely to affect the ability to detect any differences in lexical co-activation.

b) The production task appears overall more difficult than the comprehension task since participants are listening to their L2 rather than their L1 and they have to retrieve and produce a word rather than simply click on a picture. This increased cognitive load may limit the ability of the system to activate an additional language/variety.

c) In the production task, participants are explicitly asked to use two languages (English and German), whereas in the comprehension task they only need to use one (German). Perhaps having to use two languages somewhat limits the ability of the system to activate a third language/variety.

d) In the comprehension task, all competitors overlap at onset with the auditory targets; however, in the production task, for some trials competitors overlap at onset both with the auditory stimulus and with the production target (e.g., Ball [comp] – balcony [aud. stim.] – Balcon [prod. target]), whereas in other trials competitors overlap at onset only with the production target (e.g., Globus [comp] – bell [aud. stim.] – Glocke [prod. target]). Having an auditory stimulus that mismatches at onset the other two items may lead to weaker phonological activation of the critical onset (Glo- in this case). Given this, perhaps the authors could split the items based on whether the German item matches at onset its English translation and see if a difference emerges.

Minor points:

1. Lines 97-99: “Bivarietal speakers show patterns comparable to those observed in bilinguals switching between languages when asked to complete a switching task […]”: Even though there is a citation, the authors do not explicitly mention what patterns they refer to here. It would be helpful if they could briefly explain what patterns they refer to in the text.

2. It is not entirely clear what the main take-away message should be after reading the “Implications of bivarietalism” section. If the authors could add a short sentence at the end to briefly spell out what the reader should take away, that would be very useful.

3. Table 1: I understand that these scores based on the LEAP-Q; however, they should be transparent enough for a reader who is not familiar with this tool to be able to extract the relevant info directly from the table. For example, does a score of 5 reflect a 100% score? Also, what does the fluency score reflect? Finally, is there a reason why Swiss German is not included in this table?

4. How many trials were there in each VWP task? Were all four items in each set heard/produced? (e.g., was “Knopf”, the competitor of “Knochen”, ever heard?) Approx. how long did each task take? Since there is a comparison between tasks, knowing about any differences between them would be helpful.

5. The amount of phonological overlap between competitors is reported in phonemes; however, perhaps it would also be useful to have this info in milliseconds as well (i.e., approx. how many milliseconds post-stimulus-onset a target diverges from its competitor). This could be helpful when interpreting the timing of the effects.

6. Figures 1 and 2 are very helpful.

7. The section “Task 2: The production task” is a bit unclear at some points:

a) Line 311: “participants heard the target embedded in an English sentence”. You mean the English translation of the target, right?

b) Even though Figure 3 is very helpful, the corresponding info related to the task is not provided in the text in sufficient detail (as it is for the comprehension task).

c) Lines 325-237: “[…] onset of the target word in the audio stimulus came 2000 ms after stimulus onset and 1000 ms before image onset”. Based on the info provided in Figure 3, target onset came after the images. So, do you mean here 1000 ms *after* image onset?

d) I assume that participants gave their verbal responses via a microphone. This info should be included in the text.

e) How were production RTs extracted? Was duration of production extracted as well? Or only response onset?

8. How were interest areas defined for the analyses of fixation data?

9. One suggestion is to report average RTs (in addition to the lower/upper bounds currently reported) and move the RT and accuracy results earlier (i.e., before the fixation results). This would allow the reader to get an idea of the time-frame of a typical trial before moving to the “juicier” fixation data”.

10. Using the same scale for the y axes across Figures 3 and 4 would make it easier for the reader to compare fixation patterns across tasks.

Recommendation:

Overall, this is a very interesting and well-conducted study that addresses an interesting set of questions. However, I am afraid I cannot recommend it for publication in its current form. The three following issues need to be addressed: 1. Present the analytical approach and results in a clearer and more transparent way and add a clear description of how the corresponding parameter should be interpreted in terms of the underlying cognitive process(es) that it may reflect (e.g., slope can be taken to reflect speed of lexical activation); 2. Flesh out the results on the effect of SI experience on inter-/intra-dialectal co-activation; 3. Acknowledge alternative interpretations for the null effects in the production task and adjust the corresponding discussion accordingly. Hopefully, these points should be easily addressed.

Effie Kapnoula

References

Kapnoula, E. C., & McMurray, B. (2016). Training alters the resolution of lexical interference: Evidence for plasticity of competition and inhibition. J. Exp. Psychol. Gen., 145(1), 8–30.

6. PLOS authors have the option to publish the peer review history of their article (what does this mean?). If published, this will include your full peer review and any attached files.

Reviewer #1: No

Reviewer #2: **Yes: **Effie Kapnoula

---

## [Author Response · Author response to Decision Letter 0]

22 Mar 2023

Laura Keller, Ph.D.

Faculty of Translation and Interpreting

University of Geneva

Avenue du Pont-d’Arve 40

CH-1205 Geneva

Nicola Molinaro, Ph.D.

Academic Editor

PLOS ONE

Geneva, 16th March 2023

Dear Dr. Molinaro,

Thank you very much getting back to us with reviews of our manuscript so quickly. We very much appreciate the constructive way in which the aspects to improve were addressed and would like to thank the reviewers for their thoughtful and helpful guidance. We would like to hereby submit a revised version of our manuscript in which we addressed the comments made by the two reviewers. 

For easier traceability, we elaborate on how we addressed the suggestions made in detail below in black with direct reference to the individual requests printed in blue (please refer to our rebuttal letter submitted as a PDF for easier readability), followed by an overview of supporting documents added. 

FEEDBACK REVIEWER 1 

Major issues and suggestions:

1. The title very ambitiously touts simultaneous interpreting, but none of the tasks, except one, actually involve any kind of simultaneous interpreting. I’d go for something like “simultaneous interpreters”.

Thank you for this suggestion. Indeed, the tasks we ended up using are quite far removed from an actual simultaneous interpreting setting. We moved away from a more ecologically valid design to allow us to control several variables we would otherwise have been unable to account for in our analyses. We changed the title to “Unpacking the multilingualism continuum: An investigation of language variety co-activation in simultaneous interpreters” to clarify that we indeed focused on the impact of our participants’ specific profile.

2. The relevance of simultaneous interpreting (SI) or relevance thereof in co-activation could be better stated, especially because the paper proposes a specific hypothesis that is difficult to connect with the short description of SI. The main idea seems to be that co-activation is reduced to offset higher cognitive demands (comprehension higher vs. production lower, end of p. 7), but that experience in SI may again enhance co-activation (non-interpreter lower vs. interpreters higher, top of p. 8). There is some doubt in the mind of this reviewer because in lines 171-173, the authors write that “experience with SI may alter the degree of co-activation of task-irrelevant varieties due to high cognitive demands of the primary task.” The verb “alter” is not specific about the orientation of the trend and the logical conclusion readers now draw from the framing of the sentence is that co-activation would be lower in interpreters. Production is more cognitively demanding than comprehension, so activation is lower in production; ergo if SI is more cognitively demanding, activation is lower in interpreting. Or is the whole argument based on the idea that experience leads to automation, reducing cognitive demands on the interpreting task, leading to higher co-activation? If that’s the case, the missing link of automation should be included. Either way, the wording on top of p. 8 should be “altered” to make it less contradictory.

Indeed, the phrasing initially provided is not very clear — the rationale we were following was that co-activation would be less measurable in interpreters as resources would be ‘used up’ by the SI task, leaving less room for co-activation. We attempted to come up with clearer language to convey the message, while trying to avoid expanding on SI too much as to strike a good balance between the different factors under consideration and bearing in mind the overall length of the paper.

3. Interpreters’ awareness of register differences is hypothesised to account for a higher degree of co-activation of the standard German competitor. That seems fair, but it was not the focus of the research. It is therefore better to reserve the topic of register to the discussion, in stead of smuggling it into the introductory part on SI.

Thank you for pointing this out, it is indeed a fair point. The initial starting point of our considerations was that a register could potentially be viewed as a type of language variety and the original hypothesis was that independently of sociological aspects, different varieties, whatever their distinctive features, would be processed alike. However, as the project evolved over time, it is true that the focus moved away from register varieties to the more easily identifiable language variety of Swiss German.

4. The discussion could also perhaps cover an important difference between the two tasks, namely that the first task was a combination of audio input and motor response (selecting and clicking), while the second task was a combination of audio input and articulatory response. Eye movement being a motor response, the first task perhaps more easily associates with another motor response than the second?

Thank you for this helpful observation, we included it in the discussion section as follows:

It may be important to consider that the comprehension task required the processing of an audio input followed by a motor response (selecting the image corresponding to the audio input, moving the mouse cursor and clicking on that image), while the production task required the processing of an L2 audio input, the translation of the whole input (interpreter variant) or the sentence-final target word (non-interpreter variant) into the participants’ L1 as well as the articulation of the translation. This difference in the nature of the experimental tasks – the comprehension task requiring two motor responses (eye and hand movements) vs. production requiring a motor and a verbal response (eye movements and articulation of the translated sentence or target-object name), but also in comparison with the production studies, which previously found evidence for co-activation during production, but that did not comprise a language-transfer element [59, 67-69, 81]. As one reviewer helpfully suggested, it is possible that the comprehension task allowed for an easier association of responses of the same type than the second task that associated distinct response types. We indeed cannot exclude that co-activation effects lie hidden behind a more cumbersome response association for the production task in an overtly multilingual setting.

The following minor issues flagged by Reviewer 1 were addressed directly in the text of the revised paper. 

• L 112: “Unlike monolingual communication” : the authors just stated that co-activation is constant.

• L 163: “language” > “languages”

• Table 1, General, Age-mean (end of line): G3 should probably be G4

• L 253-254: “without a strong regional affiliation”: how did the authors check that? 

For this purpose we consulted the Schweizerische Idiotikon (available also online), which is the reference work on Swiss German.

• L 255 sq: the whole idea of disambiguation is unclear. 

We rephrased this passage and hope we were able to make it clearer. The idea between making the more ambiguous candidate the target was to lower naming ambiguity for the phonological competitor.

• L 333 ‘to look’ at > ‘to look at’

• Figure 3: impossible to read. The original picture that can be downloaded through the link is fine, but the one that comes with the paper is unclear. 

We are not sure what image this comment refers to, we shall provide the file again when uploading the revised paper.

• L 482: if time-out was set at 3000 ms (L 298), how can the upper bound be 3290? Participants clicked on an image they no longer saw?

Yes, indeed, the new trial started 1000ms after the images of the previous trial disappeared, therefore clicks made after the images of the ongoing triial disappeared and before the new trial started were still registered and the corresponding data for the correct trials analysed.

• L 546 sq: it is misleading to write that there are no significant signals, as the Table shows that there are. In the discussion the data are correctly represented, but it should be done here too. 

Thank you for pointing this out, we rephrased the text to better reflect the content of Table 4 as follows:

In window 1, a significant difference in terms of attracting visual attention was noted for the group of monovarietal interpreters in their task variant, however against expectations, the Standard German competitor was attended to significantly less than the distractors. Both non-interpreter groups paid significantly more attention to the Swiss German competitors during the same time window, With the exception of bivarietal non-interpreters attending significantly more to the Standard German competitors during time window 2, neither competitor type attracted significantly more visual attention than the distractors in the time spans analysed, as the results reported in Table 4 show: […].

• L 655 sq: is a bit of an exaggeration. Only bivarietal non-interpreters come close to t≥2.

That is indeed true. We deleted the sentence to avoid the risk of misleading readers.

FEEDBACK REVIEWER 2 

Major issues and suggestions:

1. The authors do a great job at reporting details of their analytical approach; however, there is some important information missing. The authors use growth curve analyses (GCA) and specifically a third-order (cubic) polynomial. My understanding is that this type of analysis produces results related to: an intercept, a linear, a quadratic, and a cubic term. However, from the manuscript it is not clear which parameter(s) the results correspond to (with the exception of one result specified as a cubic term effect in line 465). If I had to guess, I would say that the rest of the results correspond to the intercept, but in that case, it is not clear what the advantage is of using a dynamic analytical approach such as GCA instead of a traditional fixation probability measure. In any case, I am not suggesting that the authors change their analytical approach; I simply ask for some more clarity and transparency. For example, the authors could report the full models and their outputs. In addition, it would help to have some clear information as to the authors' interpretation of the corresponding parameter (i.e., the cognitive process reflected in that parameter).

Thank you for this thoughtful comment. It is indeed correct that the results reported correspond to the intercept, except for the cubic term mentioned explicitly, because it stuck out as its value reached significance. As mentioned in the text, we interpreted that value as an indicator of lower fixation proportions on cross-variety competitors in the bivarietal interpreter group, as it pointed to a shallower curvature. We opted for the dynamic analytical approach presented in our paper rather than a static fixation probability measure, because previous studies investigating activation levels (Ito et al., 2018 in particular, which we found especially helpful as LK was able to address some of her analysis questions to the first author of that paper) compared the two approaches and was found that they both yielded the same results, but that the GCA that includes a linear, quadratic and cubic time term allowed to capture non-linear changes of fixation proportions over time as well as interactions of participant group and conditions without multiple comparisons.

2. It is not clear what the overall conclusion is regarding the effect of experience with simultaneous interpreting on inter-/intra-dialectal co-activation. On the one hand, the authors conclude that “contrary to initial assumptions, SI training and professional practice do not affect co-activation patterns for comprehension”, but later on they mention that “bivarietal non-interpreters fixated both competitor types to the same extent, while bivarietal interpreters fixated the same-variety competitor significantly more than the cross-variety competitor”. I understand that the data are a bit contradictory in that respect; however, I think that this is an important point that deserves to be fleshed out a bit more. Related to this, I found it surprising that the authors do not come back to the significant effect on the cubic term (see lines 464-467). Looking at Figure 3 with this effect in mind, I can’t help but notice that bivarietal interpreters not only seem to be more distracted by same dialect competitors, but they also seem to do a much better job at resolving phonological competition: competitor looks for interpreters go up to .17-.20 and they manage to get them down to .0-.025 by the end of the trial, while for non-interpreters, competitor looks go up to only .125 and still they don’t manage to get them as low as interpreters. Interestingly, this pattern is very much in line with work reported by Kapnoula & McMurray (2016), in which we saw that a) more frequent co-activation leads to more efficient competition resolution and b) TRACE simulations showed that higher/stronger competition is accompanied by stronger inhibition of the competitor item. I am not sure how the authors could look a bit more closely into this pattern and compare competition resolution across participant groups using GCA, but if this pattern were to be backed up by stats, then one could say that SI *can* in fact give interpreters an advantage, not in terms of avoiding co-activation of task-irrelevant languages / language varieties, but in terms of resolving phonological competition more effectively.

Many thanks for these very helpful suggestions and for drawing our attention to the Kapnoula and McMurray (2016) study that provided valuable guidance on further interpretation potential of the data gathered for the study we report in our paper. Unfortunately, we were unable to corroborate a possible processing advantage stemming from efficiency gains due to increased exposure to co-activation, but picked up on that thought to enrich our discussion.

3. Finding no evidence for co-activation in production should be interpreted with more caution, given that there are a few reasons why this could be the case:

a) First and foremost, in the production task, participants were not required to click on the target. This means that they could perform the entire task without ever having to look at the images on the screen. This is likely to affect the ability to detect any differences in lexical co-activation.

This is correct. The participants were instructed to look at the screen and the experimenter reminded them of the instructions when it became apparent during the experiment that the participants’ gaze was no longer on the screen – some participants were also tempted to close their eyes to enhance concentration, which had to be monitored. The individual trial files were also screened prior to analysis to make sure fixations were registered.

b) The production task appears overall more difficult than the comprehension task since participants are listening to their L2 rather than their L1 and they have to retrieve and produce a word rather than simply click on a picture. This increased cognitive load may limit the ability of the system to activate an additional language/variety.

c) In the production task, participants are explicitly asked to use two languages (English and German), whereas in the comprehension task they only need to use one (German). Perhaps having to use two languages somewhat limits the ability of the system to activate a third language/variety.

There is no arguing with points b) and c), we are not in a position to exclude that task difficulty – and especially the potentially major jump in task difficulty from the comprehension to the production task – and the requirement of using two languages led to a sort of bottleneck or exhausted the available resources, potentially precluding a competitor effect from arising. Thank you very much for spelling out these additional possible scenarios that we gratefully included in our discussion.

d) In the comprehension task, all competitors overlap at onset with the auditory targets; however, in the production task, for some trials competitors overlap at onset both with the auditory stimulus and with the production target (e.g., Ball [comp] – balcony [aud. stim.] – Balcon [prod. target]), whereas in other trials competitors overlap at onset only with the production target (e.g., Globus [comp] – bell [aud. stim.] – Glocke [prod. target]). 

Having an auditory stimulus that mismatches at onset the other two items may lead to weaker phonological activation of the critical onset (Glo- in this case). Given this, perhaps the authors could split the items based on whether the German item matches at onset its English translation and see if a difference emerges.

While we took care to avoid ‘hidden competition’ from the object names in English and French for the competitors and distractors, we were not able to find a sufficient number of targets that did not share similar onsets in German and English to avoid them. In our exploratory search for a competitor effect, we did look at the subset of ‘overlapping targets’, speculating that a sort of double activation of a name in German and English would perhaps allow for a competitor effect to emerge. This was not the case however, so we abandoned that investigation.

Minor points:

1. Lines 97-99: “Bivarietal speakers show patterns comparable to those observed in bilinguals switching between languages when asked to complete a switching task […]”: Even though there is a citation, the authors do not explicitly mention what patterns they refer to here. It would be helpful if they could briefly explain what patterns they refer to in the text.

Thank you for pointing out that the information provided here was insufficient for uninitiated readers to follow the rationale. We expanded the section as follows to get the message across more clearly:

As for bilingual language processing, bivarietal language processing is not straightforward. Bivarietal speakers (here speakers of Dundonian/Standard English and Öcher Platt/Standard German) show patterns comparable to those observed in bilinguals switching between languages when asked to complete a switching task involving a dialectal variety and a Standard, i.e. the naming latencies were higher on switch than non-switch trials, and the switch costs in balanced bivarietals remained the same independently of the switching direction, while unbalanced bivarietals showed asymmetrical naming latencies, taking more time to switch from their non-dominant into their dominant variety than vice-versa [7]. In contrast, a picture-word interference task involving the same language-variety pair revealed no evidence for variety separation in the mental lexicon [38]. 

2. It is not entirely clear what the main take-away message should be after reading the “Implications of bivarietalism” section. If the authors could add a short sentence at the end to briefly spell out what the reader should take away, that would be very useful.

Thank you for that helpful pointer, we added the following paragraph to the section to clarify this point for the readers:

The patterns emerging from empirical evidence on bivarietal and bilingual processing therefore appear to be similar, with some of the data pointing to a separation in the lexicon [39, 40], supporting a hierarchical view [41-42], and some of it favouring an integrated connectionist view [43-48]. We therefore propose to investigate bivarietalism as a form of bilingualism.

3. I understand that these scores based on the LEAP-Q; however, they should be transparent enough for a reader who is not familiar with this tool to be able to extract the relevant info directly from the table. For example, does a score of 5 reflect a 100% score? 

Thank you again for providing an outside perspective on this. A score of 5 indeed reflects the highest possible score and we amended the description of the table to reflect this clearly.

Also, what does the fluency score reflect? 

Indeed, this is not very clearly phrased. Fluency here should read onset of fluency, as in at what age did the participants feel they were able to fluently express themselves in the language in question. We changed the nomenclature in the table accordingly to reflect the intended value measured more accurately.

Finally, is there a reason why Swiss German is not included in this table? Also, analysis of reduced data set (critical trials with competitor naming of 50% plus showed no change in effects

There is indeed. To add transparency to our rationale, we added the following section to the text:

Data on Swiss German is not included in Table 1 as the participants filled in the language-background questionnaire before coming into the lab for testing and the implication of Swiss German was only disclosed to them once the experiment completed. However, all bivarietal participants completed a post-hoc naming task, to make sure that the cross-variety competitors were recognised as such.

4. How many trials were there in each VWP task? Were all four items in each set heard/produced? (e.g., was “Knopf”, the competitor of “Knochen”, ever heard?) Approx. how long did each task take? Since there is a comparison between tasks, knowing about any differences between them would be helpful.

Our apologies for not stating this information clearly enough, we expanded the corresponding section to make that information explicit:

The 75 stimulus sets (25 per condition) used in the 75 critical trials participants completed contained the recurrent Standard German instructions (Bitte klicken Sie auf – Please click on), the spoken target and four black-and-white line object drawings. Every set was made up of a Standard German target, a phonological competitor (cross-variety or same-variety) and two unrelated distractors, or three unrelated distractors for the baseline. The placement of target and competitor was randomised and counterbalanced. Only the target names were used for audio input, neither the object names of the competitors nor of the distractors were heard during the experiment in either variety.

As illustrated in Fig 1, the comprehension task took about 15 min to complete (every critical trial lasted a total of maximum 4000 ms – target word onset was 2000 ms into the trial, followed by an extra 1000 ms of exposure to the screen with the images and another 1000 ms of blank screen), depending on how quickly participants clicked on the target they identified), whereby additional time was used for the practise trials as well as for the drift correction that occurred before every trial and which also introduced some variability regarding the overall length of the experimental task for the individual participants. The production also lasted about 15 min (the time line was identical as displayed in Fig 2, but the recording systematically continued for another 2000 ms after target-word onset).

5. The amount of phonological overlap between competitors is reported in phonemes; however, perhaps it would also be useful to have this info in milliseconds as well (i.e., approx. how many milliseconds post-stimulus-onset a target diverges from its competitor). This could be helpful when interpreting the timing of the effects.

This could indeed be useful information, however, as the papers we based our analyses on did not report phonological overlap in milliseconds, and we set the time window for analysis to allow us to capture the potential competitor effect as a function of visual attention allocated to the competitors, we were able to measure the divergence from target onset until full disambiguation, which we hope a sufficient basis for the claims we make in our contribution.

6. Figures 1 and 2 are very helpful.

Many thanks!

7. The section “Task 2: The production task” is a bit unclear at some points:

a) Line 311: “participants heard the target embedded in an English sentence”. You mean the English translation of the target, right?

Indeed, we modified the phrasing to disambiguate the task description as follows: 

The production-task design differed from the comprehension task in that participants heard the English translation of the target embedded in an English sentence […].

b) Even though Figure 3 is very helpful, the corresponding info related to the task is not provided in the text in sufficient detail (as it is for the comprehension task).

Thank you for pointing out that the information provided is not sufficient. We hope we were able to amend this shortcoming by adding the following text:

As can be gathered from Fig 3, all participants processed the same-variety competitors as competitors to the target, and while the monovarietal participants treated the cross-variety competitors like the other unrelated distractors, both bivarietal groups time courses indicate phonological cross-variety competition. Visually, the processing of both competitor types looks identical for the non-interpreter group, while a competitor-type distinction is visible for the interpreters, the same-variety competitors seemingly attracting more visual attention than the cross-variety competitors.

c) Lines 325-237: “[…] onset of the target word in the audio stimulus came 2000 ms after stimulus onset and 1000 ms before image onset”. Based on the info provided in Figure 3, target onset came after the images. So, do you mean here 1000 ms *after* image onset? 

Our apologies, that is correct. The order of the task elements was as follows: The trial started with the audio stimulus onset, which was followed by the image onset 1000 ms after the audio stimulus onset, which was in turn followed by the target word onset in the audio stimulus 1000ms post image onset. We corrected the text accordingly. 

d) I assume that participants gave their verbal responses via a microphone. This info should be included in the text.

Yes, that is correct. We added the following clarification to the text:

Participants articulated their verbal response according to the instructions their group received, and their responses were recorded via the microphone on the interpreter console installed in the test booth.

e) How were production RTs extracted? Was duration of production extracted as well? Or only response onset? 

RTs for the production task (trial onset to target-word onset in the participants’ production) were extracted from the audio files using a Praat script and the measures were then analysed in the same way as the RT measures for the comprehension task. Only target-word onset was measured, not the duration of the whole utterance the participants produced for each trial, as we were specifically interested in a potential co-activation effect generated by the participants planning, articulating, or reprocessing the target-object name in their L1 in the presence of visual same-variety or cross-variety competitor. To make the extraction process more transparent to the readership, we added the following sentence:

RT measures were extracted using a Praat [80] script and analysed analogously to the comprehension task to explore whether a potential conflict of resource use affected task-completion speed.

8. How were interest areas defined for the analyses of fixation data?

The interest areas “target”, “competitor” and “distractor” corresponded to frame of the corresponding images, which were all adjusted to be identical in size (168x168 pixels). The rest of the screen was defined as interest area “other” and fixations that fell into that area during the time windows analysed were not attributed to any of the images. We adapted the text to better reflect this.

9. One suggestion is to report average RTs (in addition to the lower/upper bounds currently reported) and move the RT and accuracy results earlier (i.e., before the fixation results). This would allow the reader to get an idea of the time-frame of a typical trial before moving to the “juicier” fixation data”.

Thank you for this suggestion, which we were happy to take up for the revised version of our paper.

10. Using the same scale for the y axes across Figures 3 and 4 would make it easier for the reader to compare fixation patterns across tasks.

While we fully agree that using the same scale for Figures 3 and 4 would have allowed for easier comparability between the two figures. However, given the differences between the tasks and the differences regarding the resulting time courses, we felt it would be equally sensible to privilege a comparability and readability of the individual data sets per task (i.e., making sure the comparability of the different groups in the comprehension task is provided for, and the same principle is applied to the production-task data) over aiming for a comparability of graphs between the two tasks. While this means that the scale of the plots reporting the data from the comprehension and the production tasks are not identical, the plots for the production task also encompass the time window analysed for the comprehension-task data, which still allows the readers to carry out an approximate comparison between the groups performances on the individual tasks.

Following the recommendations of Reviewer 2, we hope we were able to address their concerns regarding the clarity and transparency of the presentation of our analytical approach and the results, to expand on the results and their meaning regarding experience with SI and co-activation within and between varieties and to further elaborate on the interpretations for the null effects found in the production-task data in the discussion – a request also made by Reviewer 1.

We include the following elements in our resubmission in addition to the supporting information provided with our initial submission and hope they provide the transparency and clarity upheld by PLOS ONE:

• S4 Appendix: To avoid overcharging the body of the paper, we decided to provide the most salient model outputs requested by reviewer 2 in S4 Appendix along with the bases of the model scripts we ran in R. We hope this allows for a satisfactory compromise between greater transparency regarding the analyses conducted and the readability of our paper.

• S5 Appendix: Informed consent form. For your information, at the time the study was approved by the Ethics Committee, it was customary policy to request destruction of the data sets after 5 years of anonymous storage. The policy on this has changed in the meantime however, grant applications to the Swiss National Science Foundation for example have had to include data management plans as of October 2017 and the raw data files (that do not contain information allowing to identify individual participants) therefore continue to be stored on a server provided by SWITCH to Swiss Universities. As access to the server is password protected however, we opted to provide a minimal data set to fulfil the journal’s requirements regarding open science.

• S1 File: The approval of the study provided by the institution’s Ethics Committee. I (LK) have added the Ethics approval as a supporting file (S1). However, I have redacted the file as to not make my private address contained in it publicly available.

• S2 File: Minimal data set.

• Figures: As Reviewer 1 flagged an issue with Fig 3, we provide the figure files (Fig 1, Fig 2, Fig 3, Fig 4.1, Fig 4.2) again and hope this will resolve the issue.

We would like to thank the two reviewers for the great care they took in reading and reviewing our manuscript, which helped us improve our paper. We are aware of how time consuming that is and appreciate the thoroughness and diligence they applied to the task. 

My co-authors, Dr. Malte Viebahn, Prof. Dr. Alexis Hervais-Adelman, Prof. Dr. Kilian Seeber, and I hope that the changes made to the manuscript will allow you to consider it for publication in PLOS ONE.

Thank you for your consideration of our revised manuscript. 

Sincerely, 

Laura Keller, Ph.D.

---

## [Decision Letter · Decision Letter 1]

31 May 2023

PONE-D-22-33159R1Unpacking the multilingualism continuum: An investigation of language variety co-activation in simultaneous interpretersPLOS ONE

Dear Dr. Keller,

Thank you for submitting your manuscript to PLOS ONE. After careful consideration, we feel that it has merit but does not fully meet PLOS ONE’s publication criteria as it currently stands. Therefore, we invite you to submit a revised version of the manuscript that addresses the points raised during the review process. Both Reviewers were pleased with the changes to the Manuscript. One Reviewer raises some minor points that should be addressed before formal acceptance. 

We look forward to receiving your revised manuscript.

Kind regards,

Nicola Molinaro, Ph.D.

Academic Editor

PLOS ONE

Journal Requirements:

Reviewers' comments:

Reviewer's Responses to Questions

**Comments to the Author**

1. If the authors have adequately addressed your comments raised in a previous round of review and you feel that this manuscript is now acceptable for publication, you may indicate that here to bypass the “Comments to the Author” section, enter your conflict of interest statement in the “Confidential to Editor” section, and submit your "Accept" recommendation.

Reviewer #2: (No Response)

2. Is the manuscript technically sound, and do the data support the conclusions?

Reviewer #2: Yes

3. Has the statistical analysis been performed appropriately and rigorously? 

Reviewer #2: Yes

4. Have the authors made all data underlying the findings in their manuscript fully available?

Reviewer #2: Yes

5. Is the manuscript presented in an intelligible fashion and written in standard English?

Reviewer #2: Yes

6. Review Comments to the Author

Reviewer #2: I am very happy to see that the authors have done an excellent job at addressing my comments. I only have a few minor comments at this point.

1. In the previous round of review, I had asked the authors to clarify their interpretation of the cubic term parameter (i.e., the cognitive process reflected in that parameter). The authors kindly responded that they interpreted the significant effect of this factor "as an indicator of lower fixation proportions on cross-variety competitors in the bivarietal interpreter group, as it pointed to a shallower curvature". I thank the authors for adding this; however, it remains unclear to me how exactly this maps onto the underlying cognitive processes - that is, my question was referring to the linking hypothesis (see Allopenna et al., 1998). For example, in static analytical approaches (e.g., average fixation probability) the extracted measure is thought to reflect overall degree of lexical activation for that item, whereas in a dynamic approach such as curvefitting (Seedorff et al, 2018), slope can be interpreted as the speed at which lexical activation builds up. Based on the authors' response, my understanding is that they take the cubic term parameter to reflect the overall amount of lexical activation; however, in that case, I am not sure how this effect differs from an intercept effect (this is why I voiced my question in the previous round as to the advantage of using a dynamic analytical approach such as GCA instead of a traditional fixation probability measure). In any case, I don’t think this is highly problematic, but I think that the paper would benefit from a clearer description of what the cubic term is thought to reflect –not just in terms of fixations, but rather in terms of the underlying cognitive processes.

2. At the beginning of the “Fixation data” section, it is mentioned that “The fixation-time course observed – a divergence of visual attention starting at around 400 ms after target-word onset – corresponds to the 200 ms estimated for lexical access [78] added to 200 ms necessary to plan an eye movement [79]”. I just wanted to point out that this is perhaps a bit late, given that the typical timing of such effects is ~200-300 ms (see Allopenna et al., 1998). I’m only mentioning this, because the wording right now makes it sound as if 400 ms is the earliest this type of effect can show up, which is clearly not the case.

3. In response to a previous questions, the authors added that “RTs for the production task (trial onset to target-word onset in the participants’ production) were extracted from the audio files using a Praat script”. If this script is publically available, it would be helpful to add a link so that that an interested reader can see the details of the extraction procedure. Also, were these data checked by an experimenter?

4. There’s a typo at line 677: “more efficiently then the bivarietal non-interpreters”  “more efficiently than the bivarietal non-interpreters”

As mentioned above, these are all minor comments. Overall, I’m very happy with how the authors addressed previous comments and I’m looking forward to seeing this work published soon.

Allopenna, P. D., Magnuson, J. S., & Tanenhaus, M. K. (1998). Tracking the time course of spoken word recognition using eye movements: Evidence for continuous mapping models. Journal of memory and language, 38(4), 419-439.

Seedorff, M., Oleson, J., & McMurray, B. (2018). Detecting when timeseries differ: Using the Bootstrapped Differences of Timeseries (BDOTS) to analyze Visual World Paradigm data (and more). Journal of memory and language, 102, 55-67.

Effie Kapnoula

7. PLOS authors have the option to publish the peer review history of their article (what does this mean?). If published, this will include your full peer review and any attached files.

Reviewer #2: No

---

## [Author Response · Author response to Decision Letter 1]

17 Jul 2023

Laura Keller, Ph.D.

Faculty of Translation and Interpreting

University of Geneva

Avenue du Pont-d’Arve 40

CH-1205 Geneva

Nicola Molinaro, Ph.D.

Academic Editor

PLOS ONE

Geneva, 14th July 2023

Dear Dr. Molinaro,

Many thanks for the careful consideration our responses to the first round of reviews were given and for the quick turn-around. We would like to hereby submit a revised version of our manuscript in which we addressed the additional comments made by the Reviewer 2. 

As for our previous resubmission, we take up the individual requests printed in blue and elaborate on how we addressed the suggestions made in detail below each request in black (please refer to the PDF "Response to Reviewers" to see the formatting indicated here).

FEEDBACK REVIEWER 1 

No additional comments made.

FEEDBACK REVIEWER 2 

I am very happy to see that the authors have done an excellent job at addressing my comments. 

We would like to thank Reviewer 2 for their feedback on our submission following the first round of reviews, we really appreciate it. 

I only have a few minor comments at this point.

1. In the previous round of review, I had asked the authors to clarify their interpretation of the cubic term parameter (i.e., the cognitive process reflected in that parameter). The authors kindly responded that they interpreted the significant effect of this factor "as an indicator of lower fixation proportions on cross-variety competitors in the bivarietal interpreter group, as it pointed to a shallower curvature". I thank the authors for adding this; however, it remains unclear to me how exactly this maps onto the underlying cognitive processes - that is, my question was referring to the linking hypothesis (see Allopenna et al., 1998). For example, in static analytical approaches (e.g., average fixation probability) the extracted measure is thought to reflect overall degree of lexical activation for that item, whereas in a dynamic approach such as curvefitting (Seedorff et al, 2018), slope can be interpreted as the speed at which lexical activation builds up. Based on the authors' response, my understanding is that they take the cubic term parameter to reflect the overall amount of lexical activation; however, in that case, I am not sure how this effect differs from an intercept effect (this is why I voiced my question in the previous round as to the advantage of using a dynamic analytical approach such as GCA instead of a traditional fixation probability measure). In any case, I don’t think this is highly problematic, but I think that the paper would benefit from a clearer description of what the cubic term is thought to reflect –not just in terms of fixations, but rather in terms of the underlying cognitive processes.

Thank you for clarifying your question and our apologies for not getting to the intended point with our first attempt to address it. Indeed, the cubic term parameter was taken as an indicator of the curvature of lexical activation as expressed in visual attention. We based this on the assumption that lexical activation is not a process that can be captured in a linear manner, but that the incremental nature of language processing (and according to our hypothesis therefore also of language and language variety co-activation) would follow a sinusoidal rather than a linear or sigmoidal curvature, indicating an increase and subsequent decrease of visual attention after the disambiguation point during comprehension, rather than a continuous shift (Guedes et al., 2022). Although GCA is far from being the only valid approach to analysing visual world fixation data (see e.g. Seedorff et al., 2018), we opted for this analytical approach in an attempt to capture the up-and-down movement expected in terms of fixation proportions to the phonological competitor in a time window selected according to a theoretical rationale ahead of the analysis, based on the time estimated necessary according to evidence from the literature for lexical access and planning and executing the eye movement (Seedorf et al., 2018, point out in particular that high order polynomials did not live up to the promise of capturing these types of fluctuations well, however, we would argue that a third order polynomial not that high and the GCA still allows to jointly account for by-subject and by-item variability). We included this rationale in the manuscript (lines 673-689 in Revised Manuscript with Track Changes).

2. At the beginning of the “Fixation data” section, it is mentioned that “The fixation-time course observed – a divergence of visual attention starting at around 400 ms after target-word onset – corresponds to the 200 ms estimated for lexical access [78] added to 200 ms necessary to plan an eye movement [79]”. I just wanted to point out that this is perhaps a bit late, given that the typical timing of such effects is ~200-300 ms (see Allopenna et al., 1998). I’m only mentioning this, because the wording right now makes it sound as if 400 ms is the earliest this type of effect can show up, which is clearly not the case.

Indeed, the idea was to argue that the 400ms could be explained by adding the time spans reported in the literature for the execution of the two processes (lexical access and eye movement planning), not that this was the earliest and only moment at which such an effect could be observed in the data. We rephrased the sentence in question to introduce that missing nuance (lines 454-457).

3. In response to a previous question, the authors added that “RTs for the production task (trial onset to target-word onset in the participants’ production) were extracted from the audio files using a Praat script”. If this script is publically available, it would be helpful to add a link so that that an interested reader can see the details of the extraction procedure. Also, were these data checked by an experimenter?

The Praat script we refer to here was written by MCV and is not publicly available. However, during the verification process of the output produced, which was the basis to determine the accuracy scores, the output generated by the Praat script was checked by the experimenter (LK). The manuscript was updated to reflect the post-Praat verification step (lines 565-566).

4. There’s a typo at line 677: “more efficiently then the bivarietal non-interpreters”  “more efficiently than the bivarietal non-interpreters”

Thank you for the care taken in reading the manuscript again, we corrected the typo in the manuscript (line 702).

As mentioned above, these are all minor comments. Overall, I’m very happy with how the authors addressed previous comments and I’m looking forward to seeing this work published soon.

Again, a heartfelt thank you for taking the time to go through our attempts to address your comments so diligently, and for making the effort of including constructive criticism as well as praise in your feedback. We also really appreciated the additional references (Allopenna et al., 1998; Seedorf et al., 2018) that we were happy to include in our manuscript.

We would like to thank Reviewer 2 for going over our resubmitted manuscript one more time with such great attention and for again taking the time to address pertinent comments to help us further improve our paper. This degree of effort is exceptional, and we are grateful for the time invested in reviewing our work. 

My co-authors, Dr. Malte Viebahn, Prof. Dr. Alexis Hervais-Adelman, Prof. Dr. Kilian Seeber, and I hope that the additional changes made to the manuscript will allow you to consider it for publication in PLOS ONE.

Sincerely, 

Laura Keller, Ph.D.

---

## [Editor Report · Decision Letter 2]

20 Jul 2023

Unpacking the multilingualism continuum: An investigation of language variety co-activation in simultaneous interpreters

PONE-D-22-33159R2

Dear Dr. Keller,

We’re pleased to inform you that your manuscript has been judged scientifically suitable for publication and will be formally accepted for publication once it meets all outstanding technical requirements.

Kind regards,

Nicola Molinaro, Ph.D.

Academic Editor

PLOS ONE
---

## [Editor Report · Acceptance letter]

15 Nov 2023

PONE-D-22-33159R2 

Unpacking the multilingualism continuum: An investigation of language variety co-activation in simultaneous interpreters 

Dear Dr. Keller:

I'm pleased to inform you that your manuscript has been deemed suitable for publication in PLOS ONE. Congratulations! Your manuscript is now with our production department. 

Kind regards, 

on behalf of

Dr. Nicola Molinaro 

Academic Editor

PLOS ONE